# Loci-specific phase separation of FET fusion oncoproteins promotes gene transcription

Linyu Zuo[1], Guanwei Zhang [2], Matthew Massett[3], Jun Cheng[1], Zicong Guo[1], Liang Wang[2], Yifei Gao[2], Ru Li[2], Xu Huang [3✉], Pilong Li [2✉] & Zhi Qi [1✉]

Abnormally formed FUS/EWS/TAF15 (FET) fusion oncoproteins are essential oncogenic drivers in many human cancers. Interestingly, at the molecular level, they also form biomolecular condensates at specific loci. However, how these condensates lead to gene transcription and how features encoded in the DNA element regulate condensate formation remain unclear. Here, we develop an in vitro single-molecule assay to visualize phase separation on DNA. Using this technique, we observe that FET fusion proteins undergo phase separation at target binding loci and the phase separated condensates recruit RNA polymerase II and enhance gene transcription. Furthermore, we determine a threshold number of fusion-binding DNA elements that can enhance the formation of FET fusion protein condensates. These findings suggest that FET fusion oncoprotein promotes aberrant gene transcription through loci-specific phase separation, which may contribute to their oncogenic transformation ability in relevant cancers, such as sarcomas and leukemia.

[1] Center for Quantitative Biology, Peking-Tsinghua Center for Life Sciences, Academy for Advanced Interdisciplinary Studies, Peking University, Beijing, China. [2] Beijing Advanced Innovation Center for Structural Biology, Beijing Frontier Research Center for Biological Structure, Tsinghua University-Peking University Joint Center for Life Sciences, School of Life Sciences, Tsinghua University, Beijing, China. [3] Paul O'Gorman Leukaemia Research Centre, Institute of Cancer Sciences, MVLS, University of Glasgow, Glasgow, UK. ✉email: xu.huang@glasgow.ac.uk; pilongli@mail.tsinghua.edu.cn; zhiqi7@pku.edu.cn

Eukaryotic cells use lipid membrane separated and membraneless compartments for intracellular organization. Emerging evidence indicates that the latter are assembled via liquid–liquid phase separation (LLPS)[1–3]. LLPS is driven by multivalent interactions among modular biomacromolecules[4], and/or intrinsically disordered regions of proteins[5]. Aberrant phase separation has been postulated as the cause of certain cancers[6,7]. Recent studies have proposed that the transcriptional pieces of machinery function in part via LLPS[7–14].

FUS/EWS/TAF15 (FET) family proteins including FUS (Fused in Sarcoma), EWS (Ewing sarcoma), and TAF15 (TATA-binding protein-associated factor 15) contain a conserved N-terminal low complexity domain (LCD)[15] and a C-terminal RNA-binding domain. Chromosomal translocations can give rise to fusions between the FET LCDs and the DNA-binding domains (DBD) of transcription factors (TFs), like the E26 transforming sequence (ETS) family TFs[16], to form FET fusion proteins (FET-ETS). Most FET-ETS fusion proteins are oncogenic drivers in sarcomas and leukemia[15]. For example, the EWS-FLI1 fusion protein, which is frequently found in Ewing sarcoma, results from specific chromosomal translocations[17] and comprises EWS LCD linked to the DBD of the ETS family TF FLI1. Lessnick and coworkers showed that a single EWS-FLI1 binds monomerically to a 2× GGAA repeat[18,19]. FLI1DBD recognizes polymorphic GGAA motifs near transcription start sites of EWS-FLI1-bound genes[16], and subsequent EWS-FLI1 binding results in target gene activation. This process is also dependent on the number of these repetitive genomic elements (microsatellites)[20–22].

By nature, the FET LCD is intrinsically disordered, and it has been shown to form biomolecular condensates at a high concentration (~200 μM) in vitro[23]. However, the endogenous concentration of EWS-FLI1 was estimated to be ~200 nM[7]. At such a low endogenous concentration, in vivo, single-molecule imaging methods suggested that dynamic FET LCD–LCD interactions still can form biomolecular condensates at genomic binding sites, and these condensates might recruit RNA polymerase II (Pol II)[7]. Therefore, the prevailing hypothesis is that these condensates are essential for transcription, cause aberrant gene expression, and drive oncogenic gene expression programs in cancers[7]. However, this hypothesis still lacks direct evidence to support it.

Although many studies have focused on FET fusion oncoproteins, the basic biophysical nature of these proteins has still not been carefully explored and many questions remain. For example, how do the biomolecular condensates lead to gene transcription? What are the biological functions of the DNA binding sites in this process? To unravel these underlying mechanistic principles, we use a high-throughput single-molecule technique called DNA Curtains[24,25], in which arrays of individual DNA strands are arranged in the same orientation and visualized by total internal reflection fluorescence microscopy (TIRFM), to further investigate the role of FET fusion proteins in cancers by providing detailed biophysical evidence to link its biomolecular condensates with gene transcription in vitro.

In this study, based on the DNA Curtains approach, we further developed an in vitro single-molecule biomolecular condensate-induced transcription assay. Our data provided directly in vitro evidence to demonstrate the causality of FET fusion protein condensates and gene transcription, which has been suggested in several in vivo studies[7,26]. Furthermore, we identified a threshold number of fusion binding microsatellites which is critical for the promotion of the formation of FET fusion oncoproteins condensates, thereby leading to aberrant gene expression in malignant cells.

## Results

### In vitro droplet experiments indicate that FET fusion proteins undergo LLPS.

We first conducted in vitro droplet assays to ask whether FET fusion proteins can undergo LLPS. We purified a native FET fusion oncoprotein EWS-FLI1 (Supplementary Fig. 1a), and one model system, FUS-Gal4 (Supplementary Fig. 1b), which was designed by McKnight and coworkers[27]. As control experiments, the related fusion TF DBD, FLI1DBD, and Gal4DBD, were also purified and DNA fragments with scrambled sequences were used as negative controls (Supplementary Methods). For visualization, these proteins were labeled with an mCherry or GFP tag, respectively.

Electrophoretic mobility shift assays (EMSAs) were conducted to confirm that there is a specific DNA binding interaction between the fusion protein and DNA containing the fusion protein binding sequences (Supplementary Methods), and also the addition of a fluorescent tag did not affect the binding activity on DNA (Supplementary Fig. 2). Interestingly, the apparent shift in the bound complex as the EWS-FLI1 protein concentration increases implies that there might present conformational changes of the protein–DNA complex at higher concentrations of the fusion proteins, which would not likely occur if it were a stably bound monomer or dimer complex[28–30]. This instead suggests that the DNA is trapped in an aggregate (or possibly biomolecular condensates) formed by the fusion protein.

Consistent with the previous report[23], GFP-FUSLCD (Fig. 1b (i)) or even GFP alone (Fig. 1b(ii)) cannot form droplets when 30 μM of protein solution was used. However, even at 2 μM of FUS-Gal4 protein, we observed the formation of its condensates (Fig. 1a). They displayed liquid-like properties of spherical morphology and rapid fluorescence recovery after photobleaching (FRAP) with or without DNA (Fig. 1c–f), indicating that FUS-Gal4 underwent phase separation in vitro.

In comparison to FUS-Gal4, EWS-FLI1 condensates have the similar results. EWSLCD cannot form droplets when 30 μM of protein solution was used (Fig. 1h). EWS-FLI1 condensates also formed at a low protein concentration (Fig. 1g). Interestingly, EWS-FLI1 without DNA or with DNA containing 1× GGAA motif showed ~50% fluorescence recovery. Whereas, EWS-FLI1 with DNA containing 25× GGAA repeats only showed <30% recovery (Fig. 1 & Supplementary Methods), indicating that the number of GGAA repeats might affect the intrinsic DNA-binding property of the fusion protein and decrease the liquid-like property of EWS-FLI1 condensates.

### FET fusion proteins form condensates at the fusion TF DBD binding loci in vitro.

Next, we established whether DNA Curtains can be used to visualize in vitro biomolecular condensates of FET fusion proteins on DNA. To first investigate the binding events of EWS-FLI1 to a single locus, this 25× GGAA microsatellite sequence was cloned into Lambda DNA (Supplementary Methods). The sequence of 25× GGAA repeats is a crucial microsatellite for EWS-FLI1 in Ewing sarcoma located in the promoter region of NR0B1[16]. These DNA substrates were tethered to individual lipids in the supported lipid bilayer within a microfluidic chamber. The hydrodynamic force was used to align the DNA molecules along the leading edges of nanofabricated diffusion barriers, allowing visualization of hundreds of single DNA strands in a field-of-view using TIRFM (Fig. 2a, c(i))[24]. DNA was stained by YOYO-1.

When 100 nM mCherry-labeled EWS-FLI1 was flushed into the chamber (Fig. 2d and 'Methods'), high-intensity magenta puncta of EWS-FLI1 appeared on the locus at the location of the 25× GGAA repeats (Fig. 2d(ii) and Supplementary Movie 1). As control experiments, mCherry alone, mCherry-EWSLCD, or the

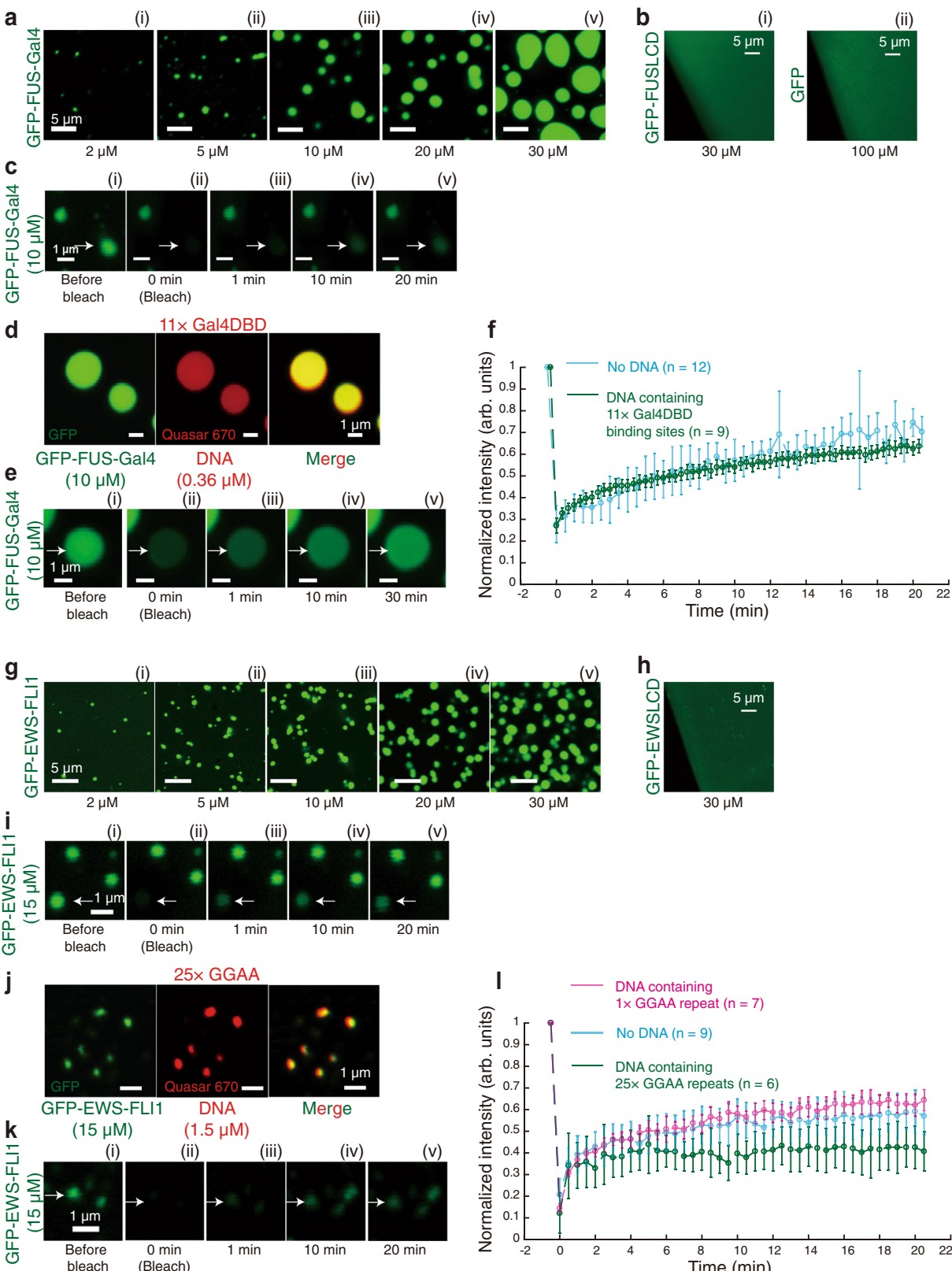

DNA-binding mutant of EWS-FLI1[31] (GFP-EWS-FLI1(R2L2)) cannot form any puncta on DNA (Fig. 2i–k).

To confirm the biomolecular condensate feature of these puncta, we examined the puncta intensities of EWS-FLI1 and the related fusion TF FLI1DBD under different protein concentrations (Fig. 2c–h). We found that the puncta intensities of EWS-FLI1 and FLI1DBD were similar at low protein concentrations, however, the puncta fluorescence intensity of EWS-FLI1 increased dramatically while that of FLI1DBD remained low when protein concentrations were greater than 250 nM. These results strongly suggested that extensive LCD interactions contributed to the cooperative multivalent interactions at the

**Fig. 1 In vitro droplet experiments indicate that FET fusion proteins undergo LLPS. a** GFP-FUS-Gal4. (i) 2 μM; (ii) 5 μM; (iii) 10 μM; (iv) 20 μM; (v) 30 μM. **b** 30 μM GFP-FUSLCD (i) and 100 μM GFP (ii). **c** FRAP experiment of 10 μM GFP-FUS-Gal4. **d** 10 μM GFP-FUS-Gal4 mixed with 0.36 μM 326-bp dsDNA. DNA contained 11× Gal4DBD binding sites and labeled with Quasar 670. **e** FRAP experiment of d. **f** FRAP curves. Blue, c; Green, e. Independent FRAP experiments were repeated: for the condition of no DNA (Blue) in c, $n = 12$; for the condition with DNA containing 11× Gal4DBD binding sites (Green) in e, $n = 9$. Error bars, mean ± s.d. **g** GFP-EWS-FLI1. (i) 2 μM; (ii) 5 μM; (iii) 10 μM; (iv) 20 μM; (v) 30 μM. **h** 30 μM GFP-FUSLCD. **i** FRAP experiment of 15 μM GFP-EWS-FLI1. **j** 15 μM GFP-EWS-FLI1 mixed with 1.5 μM 306-bp dsDNA. DNA contained 25× GGAA repeats and labeled with Quasar 670. **k** FRAP experiment of j. **l** FRAP curves. Blue, no DNA i; Green, k; Purple, DNA substrate was 1.5 μM 306-bp dsDNA containing 1× GGAA repeat. Independent FRAP experiments were repeated: for the condition with DNA containing 1× GGAA binding sites (Purple), $n = 7$; for the condition of no DNA (Blue) in i, $n = 9$; for the condition with DNA containing 25× GGAA binding sites (Green) in k, $n = 6$. Error bars, mean ± s.d. The working buffer used for the in vitro droplet assays was 40 mM Tris-HCl (pH 7.5), 150 mM KCl, 2 mM MgCl$_2$, 1 mM DTT, and 0.2 mg/ml BSA.

25× GGAA repeats, in accordance with a previous in vivo study[7]. Thus, we concluded that the high-intensity magenta puncta at the 25× GGAA microsatellite were in vitro biomolecular condensates.

Interestingly, we noticed that EWS-FLI1 condensates were also formed on other locations along with the Lambda DNA in addition to the regions containing 25× GGAA repeats (Fig. 2d (ii)). Since there were not many GGAA motifs for FLI1DBD binding along with the original wild-type Lambda DNA, we suspected that these additional condensates may be due to unspecific binding of fusion proteins. To confirm this, a fusion TF DBD possessing higher specificity to DNA was sought; the high specificity Gal4DBD was utilized by McKnight and coworkers to design a model system of FET fusion, FUS-Gal4[27].

In these experiments, we used the working buffer including 150 nM of unlabeled FUS-Gal4 to wash the chamber for 5 min and then turned down the flow to incubate for another 10 min before the flow was resumed (Fig. 3a(i)–(iv)). The DNA substrates were Lambda DNA containing 7× Gal4DBD binding sites[32] (Supplementary Methods). After the incubation, green-colored YOYO-1 puncta appeared on DNA (Fig. 3b(i)), suggesting that FUS-Gal4 was associated with the DNA through a mechanism that gave rise to regions of local high DNA-binding site concentrations.

To prove that these YOYO-1 puncta were the FUS-Gal4 condensates, we conducted a two-color experiment (Supplementary Fig. 3a–d and Supplementary Movie 2). We first injected GFP-tagged FUS-Gal4 into the chamber and observed GFP puncta appeared on DNA. We then switched to add mCherry-tagged FUS-Gal4 and found that mCherry puncta appeared and co-localized with GFP puncta. Upon loading the working buffer including YOYO-1, we further observed much higher green-color signals (Supplementary Fig. 3e). We considered the enhanced green-color signals were from YOYO-1 because there were no free GFP-tagged FUS-Gal4 molecules in the chamber at this time. We also found that the puncta can undergo fusion across different strands (Supplementary Movie 3 and Fig. 3b(iii)). The fusion events appeared 24 times for 64 pairs of DNA substrates containing puncta, suggesting that these events frequently occurred in contrast to no event observed without the addition of FUS-Gal4 (Fig. 3b(iv)). Moreover, we analyzed the position distribution of these YOYO-1 puncta, and a 1D Gaussian function was used to fit the peak position (~9060 ± 1254 nm, $N = 201$, Fig. 3b(ii)), which closely coincided with the insertion position of Gal4DBD binding sites (9124 nm). These results demonstrated that these YOYO-1 puncta were FUS-Gal4 condensates, which preferred to bind to the seven repeats of the Gal4 DNA-binding site, possessing a liquid-like property. In striking contrast, treatments with 10 nM unlabeled FUS-Gal4 (Supplementary Fig. 4a(i)) and 150 nM unlabeled Gal4DBD (Supplementary Fig. 4b(i)) failed to generate YOYO-1 puncta on DNA, confirming again that FUS LCD–LCD interactions are essential for FUS-Gal4 condensate formation, corroborating our previous results using EWS-FLI1 (Fig. 2).

When FET fusion proteins form condensates on DNA, we wondered whether DNA length can be affected. We conducted the image tracking analysis of DNA ends to measure "Shrinkage (%)", which was defined as the length change over the initial length and represented DNA compaction (Supplementary Fig. 5a)[33]. We found that FUS-Gal4 condensates only slightly compacted the DNA substrates (<10%) (Supplementary Fig. 5b), even in the case of low salt concentration (<25% for 10 mM salt) (Supplementary Fig. 5c). The real FET fusion oncoprotein EWS-FLI1 led to a similar conclusion, even in the case of higher protein concentration (<35% for 500 nM EWS-FLI1 (Supplementary Fig. 5d).

**FET fusion protein condensates recruit Pol II CTD to fusion binding motif loci in vitro.** Recent in vivo studies suggested interactions between FET fusion protein condensates and Pol II[7], and in vitro studies demonstrated interactions between FET LCD and the C-terminal domain (CTD) of Pol II[23,27]. If these YOYO-1 puncta of FUS-Gal4 were FUS-Gal4 condensates, they might be able to interact with the Pol II CTD.

To test this possibility, we first constructed a clone expressing the N-terminal heptapeptide repeat 1–26 of the human Pol II CTD[34] tagged with mCherry (termed Pol II CTD$_{N26}$-mCherry) and the protein was purified from *E. coli.* to ensure that no phosphorylation modifications were present on the protein. Vertebrate Pol II CTDs include 52 heptapeptide repeats, and the N-terminal half of the CTD is universally conserved from yeast to human[34]. Unphosphorylated Pol II CTDs can be recruited into the biomolecular condensates of FET fusion proteins while the phosphorylated CTD can dissociate from the condensates[10,27]. When we conducted an in vitro droplet assay with mixed Pol II CTD$_{N26}$-mCherry (5 μM) and GFP-FUS-Gal4 (30 μM) together, they coalesced within condensates (Fig. 3c), and FRAP experiments confirmed that the droplet possessed liquid-like property (Fig. 3d, e). We also conducted the control experiments and found no droplets formed when GFP-FUS-Gal4 was exchanged to GFP (Fig. 3f) or GFP-FUSLCD (Fig. 3g). These results confirmed that Pol II CTD$_{N26}$-mCherry can be specifically recruited into the FUS-Gal4 condensates under lower concentrations.

Next, we conducted DNA Curtains and established FUS-Gal4 condensates at the 7× Gal4DBD binding sites as in Fig. 3b, followed by injection of 1 μM Pol II CTD$_{N26}$-mCherry into the chamber for a 10 min incubation with the FUS-Gal4 condensates (Fig. 3a(iv–vi)). We observed that Pol II CTD$_{N26}$-mCherry co-localized extensively with the FUS-Gal4 puncta located at the Gal4DBD binding sites (Fig. 3h and Supplementary Movie 4). As a control experiment, Pol II CTD$_{N26}$-mCherry did not associate with DNA directly without the presence of biomolecular condensates (Supplementary Fig. 4a(ii–iv), b(ii–iv)). We further confirmed the results with EWS-FLI1 (Supplementary Fig. 6a, d, and Supplementary Movie 5), supporting that FET fusion protein condensates can recruit Pol II CTD at the target loci. As a control

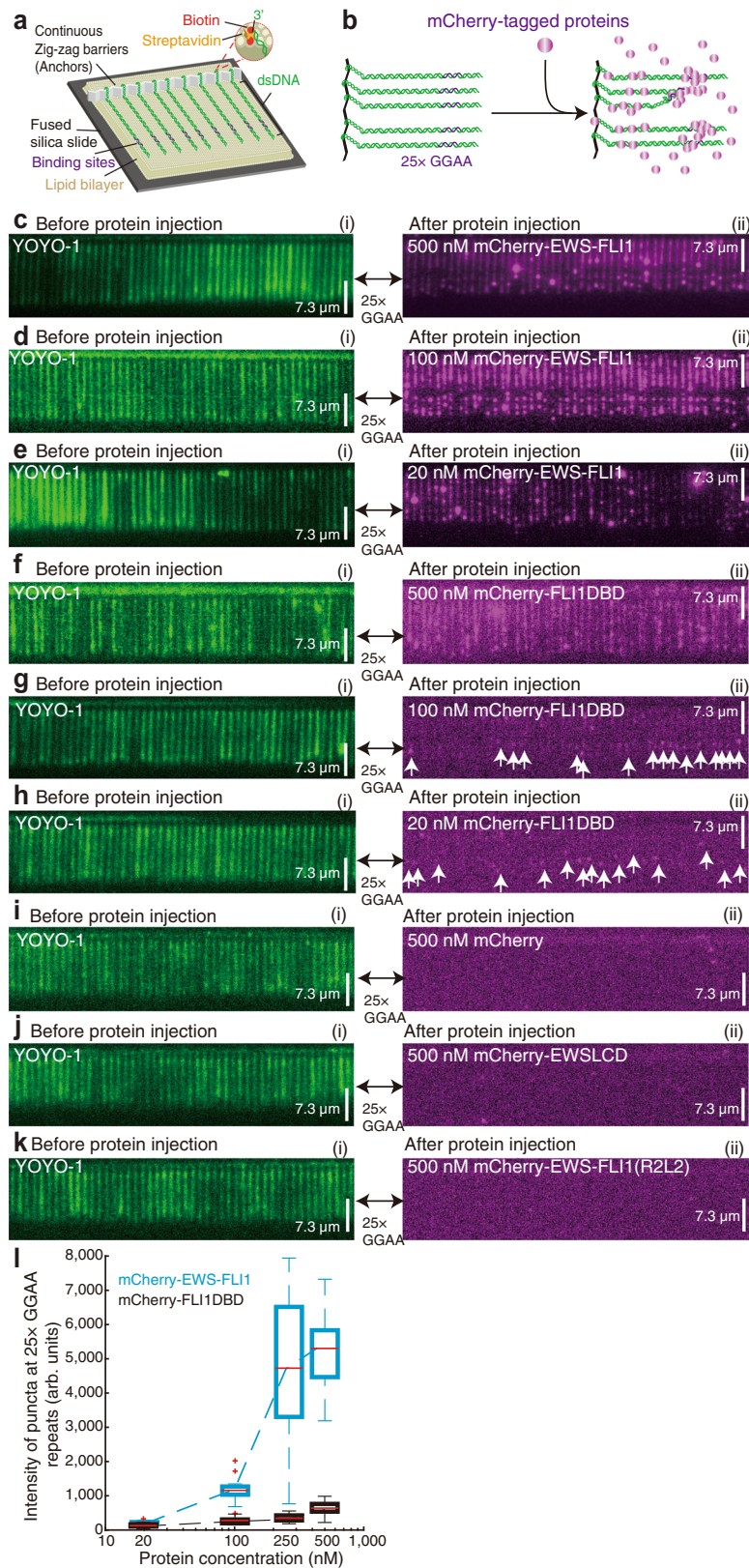

experiment, FLI1DBD cannot recruit Pol II CTD (Supplementary Fig. 6c, d).

**The recruitment of RNA polymerase by FET fusion protein condensates promotes gene transcription activity in vitro.** Next, we examined whether the recruitment of Pol II by the fusion

condensates can activate gene transcription around the target loci. Previous in vivo studies suggested a potential correlation between FET condensates and transcription activities[7,26]. However, these studies cannot provide direct evidence to prove a causal relationship. Here we sought to develop a single-molecule in vitro transcription assay on DNA Curtains to demonstrate this causality.

**Fig. 2 EWS-FLI1 forms condensates at the FLI1 binding loci in vitro. a** Schematic of DNA Curtains. **b** Strategy for detecting EWS-FLI1 on DNA Curtains. **c**–**k** Wide-field TIRFM images of DNA Curtains before (i) and after (ii) 100 nM protein injection (a time point at the 5th minute). mCherry-EWS-FLI1: 500 nM c, 100 nM d, and 20 nM e. mCherry-FLI1DBD: 500 nM f, 100 nM g, and 20 nM h. 500 nM mCherry i, 500 nM mCherry-EWSLCD j, and 500 nM mCherry-EWS-FLI1(R2L2) k. DNA substrate was YOYO-1-stained Lambda DNA containing 25× GGAA repeats. **l** Boxplot of the puncta intensities of EWS-FLI1 (cyan) and FLI1DBD (black) at the 25× GGAA repeats versus protein concentration. The total number $N$ of puncta at 25× GGAA repeats examined over one time DNA Curtain's experiment: (1) for mCherry-EWS-FLI1, 20 nM ($N = 10$), 100 nM ($N = 20$), 250 nM ($N = 30$), and 500 nM ($N = 20$); (2) for mCherry-FLI1DBD, 20 nM ($N = 20$), 100 nM ($N = 14$), 250 nM ($N = 30$), and 500 nM ($N = 25$). For the boxplot, the red bar represents median. The bottom edge of the box represents 25th percentiles, and the top is 75th percentiles. Most extreme data points are covered by the whiskers except outliers. The '+' symbol is used to represent the outliers.

As T7 RNA polymerase (T7 RNAP) possesses highly efficient activity with a single subunit structure in vitro[35], we first chose T7 RNAP for the in vitro transcription assay. After the biochemical activity of T7 RNAP was tested (Supplementary Fig. 7a), we injected T7 RNAP and NTPs into the DNA Curtains chamber containing DNA with a 1× T7 promoter (Supplementary Methods) at 37 °C for 20 min. Fluor647-tagged UTP in an NTP mixture was used to label nascent RNA transcripts (Supplementary Fig. 7b). As expected, magenta puncta appeared following the incubation (white arrows in Supplementary Fig. 7c(ii)), indicative of nascent RNA transcripts. We observed a relatively low transcription efficiency (~0.13, $N = 308$, Lane 1 in Supplementary Fig. 7g), which was calculated as the ratio of the number of nascent RNA transcript puncta to the number of DNA molecules.

To validate whether the nascent RNAs were correctly transcribed in our assay, we inserted 6× MS2 binding sequences after the T7 promoter on the Lambda DNA (Supplementary Methods), and confirmed that 3× Flag-tagged MS2 protein purified from E. coli was able to bind de novo RNA containing MS2 binding sequences (Supplementary Fig. 7d). We repeated the experiment in Supplementary Fig. 7b in which unlabeled NTPs were used followed by the injection of MS2-3× Flag labeled with Flag-Quantum dot (QD) 705 to visualize RNA transcripts (Supplementary Fig. 7e). Large quantities of magenta puncta appeared on DNA (Supplementary Fig. 7f(ii) and Supplementary Movie 6), indicating that RNAs were transcribed as expected in our in vitro transcription assays.

We noticed that the transcription efficiency of the MS2 labeling method was higher than UTP-Fluor647 labeling method (Supplementary Fig. 7g). We believe that the reason was the concentration of NTPs. The MS2 labeling method used ~mM levels of NTPs, in accordance with the previous T7 RNAP transcription protocol[36]. However, the UTP-Fluor647 labeling method used ~μM levels of NTPs because of the commercial concentration limit of UTP-Fluor647, suggesting that in vitro transcription efficiency was sensitive to the NTP concentration.

As the Pol II CTD can be recruited into FET fusion protein condensates (Fig. 3h and Supplementary Fig. 6a), we questioned if we fused T7 RNAP and Pol II $CTD_{N26}$ together, whether this fusion complex can be recruited into FUS-Gal4 or EWS-FLI1 condensates, and then the fusion T7 RNAP inside the condensates might activate gene transcriptions on DNA nearby inside the condensates. Therefore, we fused T7 RNAP to Pol II $CTD_{N26}$, forming Pol II $CTD_{N26}$-T7 RNAP. FUS-Gal4 condensates were formed at the 7× Gal4DBD binding sites on the DNA Curtains, followed by injection of Pol II $CTD_{N26}$-T7 RNAP, Pol II $CTD_{N26}$-mCherry, and NTPs including Fluor647-tagged UTP into the chamber for a 20-min incubation at 37 °C (Fig. 4a). As expected, enriched Fluor647 signals representing nascent RNA transcripts were detected within FUS-Gal4 condensates (white arrows in Fig. 4a(iii) and Supplementary Movie 7). Since there was no apparent T7 promoter included in this assay, our results strongly suggested that FUS-Gal4 condensates indeed induced

transcription. Moreover, the enriched Fluor647 signals indicated a high transcription efficiency (0.74, $N = 323$, Lane 2 in Fig. 4f).

We further conducted a set of control experiments to confirm our findings. To guarantee that the Fluor647 signals in the chamber did not come from the free UTP-Fluor647 binding to DNA or the FUS-Gal4 condensates, we repeated the experiment without the addition of Pol II $CTD_{N26}$-T7 RNAP (Fig. 4e). The result showed that there were few detectable Fluor647 signals (0.11, $N = 442$), suggesting that free UTP-Fluor647 barely binds to DNA and the FUS-Gal4 condensates. To confirm the role of FUS-Gal4 condensates, we also performed the assay without the addition of FUS-Gal4 (Fig. 4c). The resulted transcription efficiency was very low (0.008, $N = 375$, Lane 3 in Fig. 4f), suggesting that in vitro transcription in this system depends on FUS-Gal4 condensates formation. We also found that the salt concentration affected the in vitro transcription efficiency, and 150 mM KCl caused a much lower transcription efficiency than 25 mM KCl (Fig. 4b and Lane 1 in Fig. 4f). Thus, in this paper, we used 25 mM KCl for the in vitro transcription assay.

Together, these results clearly demonstrated that the causality between our model system FUS-Gal4 condensates and gene transcription. We further confirmed the results with EWS-FLI1 (Fig. 4g), supporting that the real FET fusion protein condensates can also recruit the RNA polymerase to activate gene transcription around the target loci. Moreover, in comparison to the transcription efficiency of wild-type T7 RNAP on 1× T7 promoter (Supplementary Fig. 7b, c), FUS-Gal4 condensates enhanced gene transcription, even without a T7 promoter on DNA substrates (Fig. 4a). Interestingly, using wild-type Lambda DNA (without our designed Gal4DBD binding sites) as DNA substrates in this in vitro transcription assay (Fig. 4d) provided a very low transcription efficiency (0.05, $N = 278$, Lane 4 in Fig. 4f). These data strongly suggested that the DNA binding motif might be another important factor controlling in vitro transcription. This notion promoted us to further investigate the role of DNA binding motifs in the fusion condensate-lead gene transcription.

**The number of GGAA microsatellites is highly associated with gene transcriptional regulation by FET fusion proteins.** DNA binding motifs indeed possess critical gene regulatory functions. For example, recent studies showed that targeted silencing of specific GGAA microsatellite genomic loci induced a reduction of SOX2 expression, a key gene in Ewing sarcoma, leading to tumor growth regression in vivo[20,21]. Lessnick and coworkers also found that maximal EWS-FLI1-mediated gene expression was associated with 20–26× GGAA-microsatellite-length polymorphisms[22]. These results strongly suggested a threshold feature of GGAA repeat-containing microsatellites.

We first followed the previously reported luciferase assays[22,37] to examine length polymorphisms of GGAA microsatellites. For EWS-FLI1, we found that when the length of GGAA-microsatellites increased, elevated relative luciferase activity was also observed (Fig. 5b). The increased transcriptional activity of EWS-FLI1 exhibited a dependence on a threshold number of the

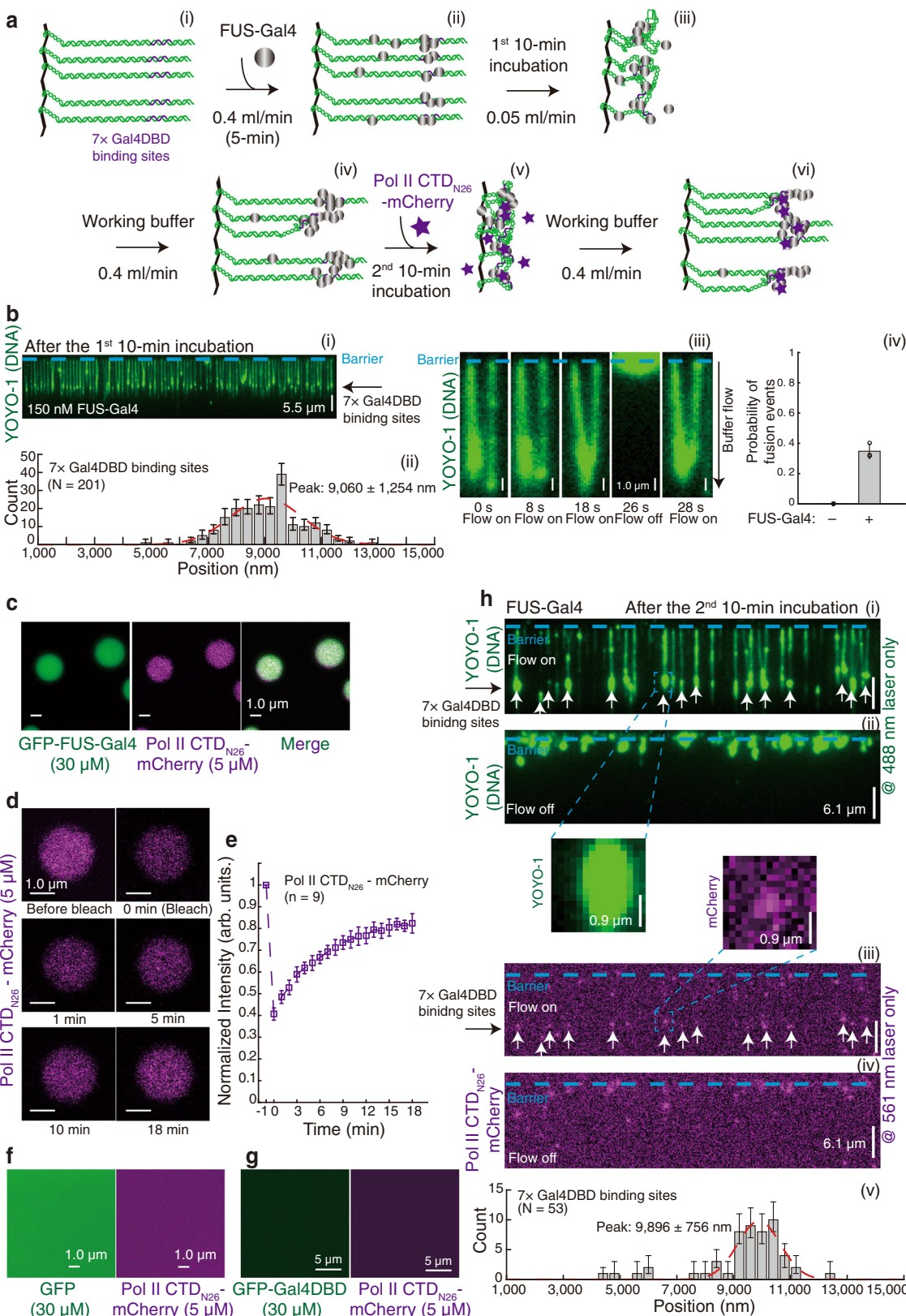

binding motifs. The Hill equation was used to quantitatively fit the threshold (~6× GGAA), suggesting that the number of GGAA repeats beyond the threshold number of repeats leads to gene transcriptional activation. FUS-Gal4 (Fig. 5a) and another important FET fusion oncoprotein FUS-ERG (Fig. 5c), which is found in Ewing sarcoma and acute myeloid leukemia (AML)[15],

shared the similar threshold feature encoded in a repeated DBD binding-motif. We also performed the control experiment without co-transfection of the fusion proteins (Fig. 5d). There was little luciferase activity observed without the fusion protein, further supporting that the fusion protein is responsible for the increased transcription of these longer GGAA repeats.

**Fig. 3 FUS-Gal4 condensates recruit Pol II CTD to fusion binding motif loci in vitro. a** Strategy for detecting FUS-Gal4 condensates on DNA and loci-specific Pol II CTD recruitment. DNA substrate was Lambda DNA containing 7× Gal4DBD binding sites. **b**(i) Wide-field TIRFM images of DNA Curtains after the 1st 10-min incubation. (ii) Position distribution of YOYO-1 puncta in (i). 400-nm bin. $N = 201$, the total YOYO-1 puncta number examined over three times DNA Curtains experiments. (iii, iv) Five panels showing time course of the fusion event by two FUS-Gal4 condensates on different DNA substrates in trans, time at bottom in seconds. Independent DNA Curtains experiments in **b**(i) were repeated three times ($n = 3$). Error bars, mean ± s.d. **c–g** In vitro droplet assays. **c–e** Pol II CTD$_{N26}$-mCherry molecules were recruited into the droplet of GFP-FUS-Gal4. **c** GFP-FUS-Gal4 (30 μM) was mixed with Pol II CTD$_{N26}$-mCherry (5 μM). **d, e** FRAP experiments. Independent FRAP experiments in (**d**) were repeated, $n = 9$. Error bars, mean ± s.d. Pol II CTD$_{N26}$-mCherry molecules mixed with GFP (30 μM) alone (**f**) or GFP-Gal4DBD (30 μM) alone (**g**). **h** Wide-field TIRFM images of FUS-Gal4 condensates and Pol II CTD$_{N26}$-mCherry after the 2nd 10-min incubation. (i, ii) only the 488-nm laser on and flow on (i) / off (ii); (iii, iv) only the 561-nm laser on and flow on (iii) / off (iv); White arrows pointed to puncta of Pol II CTD$_{N26}$-mCherry colocalized with FUS-Gal4 condensates. Inserts were representative colocalized puncta. (v) Position distribution of magenta puncta in (ii). 400-nm bin. $N = 53$, the total colocalized puncta number examined over three-time DNA Curtains experiments ($n = 3$). Error bars in **b**(ii) and **h**(v) were obtained through the bootstrap analysis. For any normally distributed dataset, 68.27% of the values lie within one standard deviation of the mean, therefore our choice of 70% confidence intervals for the bootstrapped data provides a close approximation to expectations for one standard deviation from the mean. The data were fitted with a Gaussian function (red dash line). Peak center: **b**(ii), 9060 ± 1254 nm; **f**(iv), 9896 ± 756 nm. The errors represented 95% confidence intervals obtained through Gaussian function fitting.

Furthermore, we conducted a bioinformatics analysis of chromatin immunoprecipitation sequencing (ChIP-seq) data (NCBI, GEO: GSE99959)[38] to characterize the promoter-like EWS-FLI1-bound microsatellites in Ewing sarcoma cell line A673. We adopted both consecutive motif (in which the linker sequence between two GGAA motifs was equal or less than 1-bp) and total motif (in which the linker sequence between two GGAA motifs was equal or less than 20-bp) analyses previously defined in[38] to record the distribution of GGAA microsatellites in the genome. Both the fold enrichment of EWS-FLI1 in promotor regions and the related gene expression exhibited a threshold number of GGAA repeats regardless of which types of motifs were recorded (Supplementary Fig. 8a–d). These results were also shown previously by Lessnick and coworkers[18,19,37,38]. These data and our results from luciferase assays above (Fig. 5) concordantly demonstrated that in the in vivo setting, there exists a threshold number of GGAA microsatellites highly associated with the gene transcriptional regulation by FET fusion proteins.

**DNA binding motifs beyond a threshold number drive the formation of FET fusion protein condensates**. We then sought to understand the biophysical basis of these particular DNA binding motifs in vitro. We first cloned 1×, 5×, 7×, or 11× Gal4DBD binding sites into Lambda DNA (Supplementary Methods), and examined the behavior of FUS-Gal4 proteins on these engineered DNA substrates with DNA Curtains. We observed many of the puncta co-localized with the Gal4DBD binding sites (Supplementary Fig. 9a, b). For example, the positional distribution of 7× Gal4DBD binding sites was fitted with a 1D Gaussian function (Supplementary Fig. 9b), which closely coincided with the insertion position of Gal4DBD binding sites (9124 nm). Co-localization only occurred at the repeats of the Gal4DBD binding sites ≥7 (Supplementary Fig. 9), indicative of a threshold number of Gal4DBD binding sites.

Similarly, we next cloned 3×, 5×, 7×, 9×, 13×, or 25× GGAA repeats into Lambda DNA and examined EWS-FLI1 binding. Using the same method carried out in Fig. 2b, we loaded 500 nM mCherry-EWS-FLI1 into the chamber, followed by a 2-min wash of the chamber with the working buffer to flush away all unbound proteins from the chamber. We then turned off the flow for a 10-min incubation. Afterward, we turned on the flow to linearize DNA substrates (Fig. 6a). If the threshold feature from the in vivo experiments (Fig. 5 & Supplementary Fig. 8) is consistent in vitro, we would expect to see an individual magenta punctum present at the regions containing a number of repeats over the threshold number.

As expected, when using the engineered lambda DNA containing 25× GGAA repeats, we indeed observed magenta

puncta appearing on each string of DNA (Fig. 6b(i)), implicating that a threshold number of GGAA repeats is required for mCherry-EWS-FLI1 binding. Unexpectedly, in addition to the predicted magenta punctum positioned at 25× GGAA repeat region, we also observed many extra magenta puncta (Fig. 6b(ii–v)), implicating that EWS-FLI1 condensates may bind at other specific loci different from the purposely engineered region (containing 25× GGAA repeats). Moreover, when we repeated the experiment with DNA containing an insert of a different number of GGAA repeats from 13× (Fig. 6c(i)), 9× (Fig. 6d(i)), 7× (Fig. 6e(i)), 5× (Fig. 6f(i)), 3× (Fig. 6g(i)) GGAA repeats, or even with the wild-type Lambda DNA (Fig. 6h(i)), the resulting images also displayed similar patterns to those observed with 25× GGAA repeats in Fig. 6b(i).

To investigate these binding patterns using ChIP-seq data analyses (Supplementary Fig. 8a, b), we conducted both consecutive motif (in which the linker sequence between two GGAA motifs was equal or less than 1-bp) and total motif (in which the linker sequence between two GGAA motifs was equal or less than 20-bp) analyses to record the distribution of GGAA microsatellites on these engineered Lambda DNA substrates. If we assumed that the threshold was ~3× GGAA repeats (dashed red lines in Fig. 6), the consecutive motif analysis for Lambda DNA containing 25× GGAA repeats can only predict a single punctum present at the region containing these 25× GGAA repeats (Fig. 6b(vi)). However, the total motif analysis predicted more puncta appearing on DNA because multiple regions on the DNA included a number of total motifs that are greater than the threshold number of GGAA repeats (Fig. 6b(vii)). Similarly, the total motif analysis for Lambda DNA containing GGAA repeats from 13 to 3 (Fig. 6c(ii), d(ii), e(ii), f(ii), g(ii), and h(ii)) and wild-type Lambda DNA (Fig. 6h(ii)) can also explain the presence of extra magenta puncta, representing mCherry-EWS-FLI1 condensates.

## Discussion

In this study, we utilized a single-molecule imaging technique, DNA Curtains, to fully characterize the biophysical nature of FET fusion protein-led condensates in vitro, providing the first direct evidence linking these condensates with gene transcriptional activation (Fig. 7). Once the specific target loci (total motif number ≥ threshold number) appear, the threshold number of binding sites facilitates the collection of local FET fusion oncoprotein concentration required to form biomolecular condensates on the target loci, which leads to the recruitment of Pol II to the target loci. Finally, our developed in vitro single-molecule biomolecular condensate-enhanced transcription assay confirms that the recruited Pol II molecules indeed activate gene transcription,

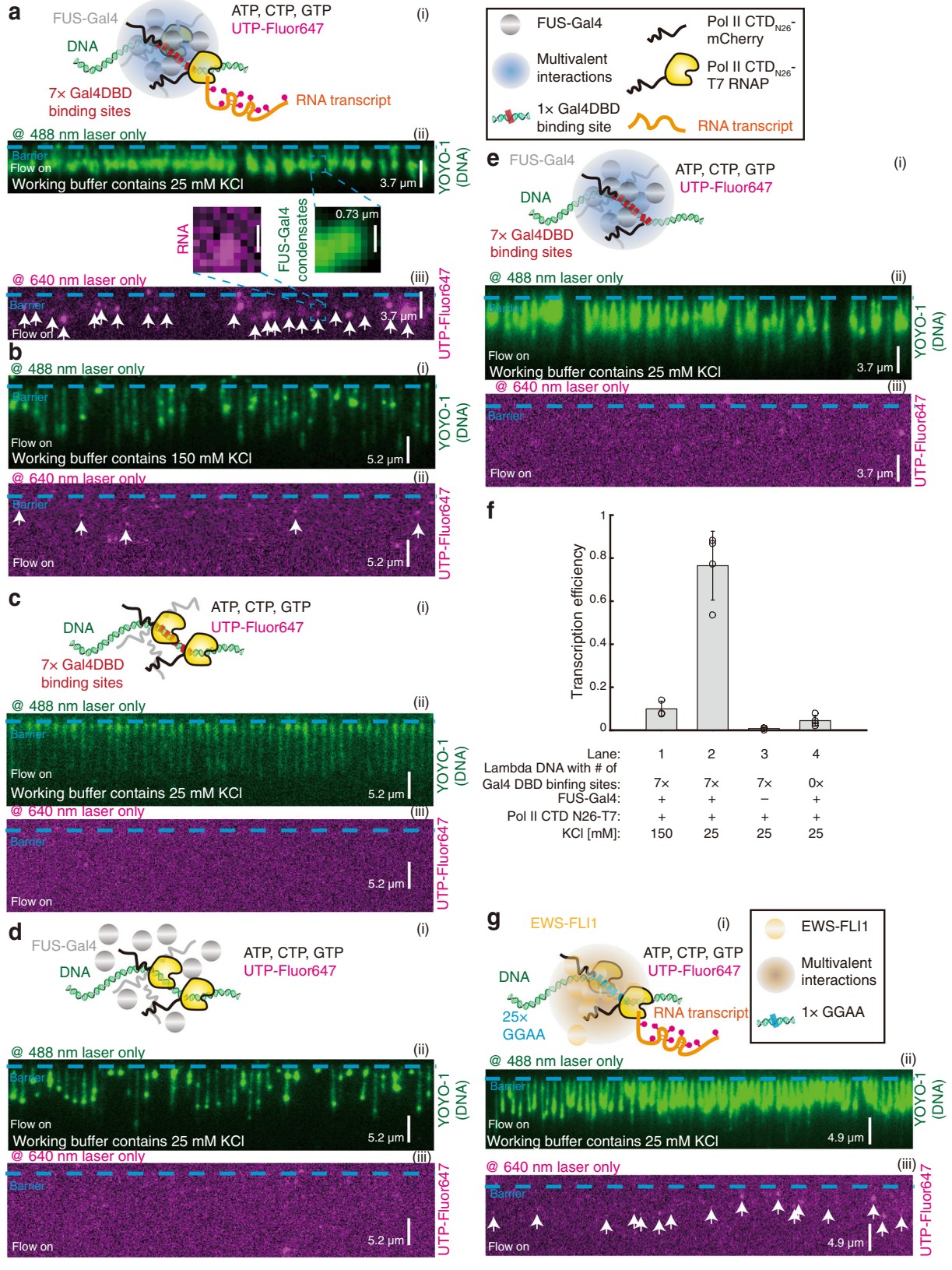

and hence possibly exerts aberrant transactivation in cancer patient cells[16,37].

In order to further clarify the contributions of protein–protein and protein–DNA interactions to the FET fusion protein condensates formation, we designed the control experiments to interrogate the phase behavior of FUS-Gal4 in the presence of

oligos comprising the Gal4DBD binding sites (Supplementary Fig. 10). GFP-FUS-Gal4 formed small droplets at 8 μM (Supplementary Fig. 10a(i)). Mixing GFP-FUS-Gal4 with DNA including varying numbers of Gal4DBD binding sites resulted in larger droplets (Supplementary Fig. 10a, d). The Gal4DBD alone did not form droplets without (Supplementary Fig. 10b) or with

**Fig. 4 The recruitment of RNA polymerase by FET fusion protein condensates promotes gene transcription activity in vitro. a** Schematic of in vitro single-molecule biomolecular condensate-induced transcription assay for FUS-Gal4 (i). Wide-field TIRFM images of nascent RNA transcripts (iii) colocalized with FUS-Gal4 condensates (ii) after incubation. White arrows in (iii) confirmed the labeled punctum was on DNA. The insert was representative colocalized puncta. DNA substrates were Lambda DNA containing 7× Gal4DBD binding sites. The working buffer was 40 mM Tris-HCl (pH 7.5), 25 mM KCl, and 2 mM MgCl$_2$, 1 mM DTT, and 0.2 mg/ml BSA. **b** The control experiment of a: the working buffer containing 150 mM KCl in this transcription assay. **c** The control experiment of a: no FUS-Gal4. **d** The control experiment of a: DNA substrates were wild-type Lambda DNA (no 7× Gal4DBD binding sites). **e** The control experiment of a: no Pol II CTD$_{N26}$-T7 RNAP. **f** Transcription efficiency for different experimental conditions for FUS-Gal4 from a to d. Independent DNA Curtains experiments were repeated: $n = 3$ for a, $n = 4$ for b to d. Error bars, mean ± s.d. **g** Schematic of in vitro single-molecule biomolecular condensate-induced transcription assay for EWS-FLI1 (i). Wide-field TIRFM images of nascent RNA transcripts (iii) colocalized with EWS-FLI1 condensates (ii) after incubation. DNA substrates were Lambda DNA containing 25× GGAA repeats. White arrows in (iii) confirmed the labeled punctum was on DNA. The working buffer contained 25 mM KCl in a, c, d, e, and g.

DNA (Supplementary Fig. 10c). Phase separation diagrams of protein and DNA (Supplementary Fig. 10e–g) revealed: (i) the presence of DNA promoted the FUS-Gal4 condensates by lowering the critical concentration for FUS-Gal4 phase separation; and (ii) the presence of more Gal4DBD binding sites on the DNA resulted in even lower critical concentration of FUS-Gal4. The phase boundary showed a strong synergy between the concentration of DNA binding motifs and the concentration of FET fusion protein. These results strongly suggested that the FET fusion protein condensates can form around the DNA leveraging a combination of homotypic interactions (presumably involving the LCD and DBD of FET fusion protein) and heterotypic interactions (mainly involving the DBD and DNA) whereby the DNA weaves through condensates. As we already confirmed GFP-FUSLCD (Supplementary Fig. 2i) cannot bind to the 1× Gal4DBD sequence and also the control sequence, the LCD and DNA cannot form the heterotypic interactions mentioned above.

Interestingly, the droplet fluorescence also exhibited an abrupt jump when the DNA contained ≥7 Gal4DBD binding sites (Supplementary Fig. 10a, d), suggesting a threshold of 7× Gal4DBD binding sites enhanced FUS-Gal4 condensates. This result matches well with the results of the luciferase assay (Fig. 5a) and DNA Curtains (Supplementary Fig. 9).

DNA Curtains have been used in previous biomolecular condensate studies. Human heterochromatin protein 1α (HP1α)[39] and vernalization 1 (VRN1)[33] induce the gene silencing in cells and completely compact DNA on DNA Curtains upon LLPS. However, our observations for FUS-Gal4 (Supplementary Fig. 5a–c) and EWS-FLI1 condensates (Supplementary Fig. 5d) suggested that FET fusion oncoproteins may have different mechanisms of biomolecular condensation from HP1α and VRN1 as far as their interactions with DNA are concerned. The distinction in their DNA compaction capabilities may be consistent with their different roles in transcription regulation. It is well known that HP1α requires extensive DNA compaction to form heterochromatin, mediating transcription repression[39]. VRN1 has a similar transcriptional repression function by suppressing the expression of the flowering repressor gene Flowering Locus C (FLC)[33]. However, FET fusion oncoproteins require uncompacted DNA to implement aberrant transactivation activities[7,15,16,27].

FET fusion proteins undergo LLPS in vitro at ~10 μM, which is much lower than the concentration needed for FET LCD (Fig. 1), strongly suggesting the fusion TF DBD may play an important role in FET fusion protein condensates formation. Recent studies of FUS found the interactions between the LCD and RNA-binding domain (RBD) was responsible for LLPS[40,41]. Moreover, the different behaviors of EWS-FLI1 and FLI1DBD in Fig. 2l suggest that not only LCD–LCD interactions[7] but also LCD–DBD interactions may contribute to the EWS-FLI1 condensate formation. Lessnick and coworkers indeed confirmed the LCD–DBD interactions[19]. Thus, there should be some commonalities between the DBDs in FET fusion proteins and the RBDs in FET

proteins[40], and it is worth exploring in the future whether the interactions between FET LCD and the fusion TF DBD can also drive the formation of biomolecular condensates.

Lessnick and coworkers have also shown that consecutive motifs are more highly bound by EWS-FLI1 by ChIP-seq data analyses[38], and even a change of a single base pair can generate a new string of consecutive motifs[42]. These in vivo pieces of evidence indicate that consecutive motifs carry real biological significance. However, our in vitro results showed that the total number of GGAA motifs is better to explain the biomolecular condensate formation on DNA Curtains. Thus, the relationship between total motifs and consecutive motifs is another interesting topic for the future study.

However, the presence of total motifs in addition to the consecutive motifs impels us to consider the potential interplay between the FET fusion oncoprotein condensates and genomic DNA at a global level. Genomic DNA may act as a scaffold for the formation of FET fusion oncoprotein condensates at specific loci. The definition of total motif strongly suggests that GGAA repeats do not need to be spatially connected, and the GGAA repeat density in the genome may be more important to regulate the formation of condensates as enhancer-like microsatellites as suggested by Lessnick and coworkers[38]. This is further supported by the observation that the enhancer density affects transcriptional condensates formation[14]. Vice versa, the FET fusion oncoprotein condensates may contribute to reshaping of genomic DNA through interactions with spatial GGAA repeats, exemplified by the transcriptional condensates of TFs and cofactors which can also influence chromosome conformation[43].

Our developed in vitro single-molecule biomolecular condensate-enhanced transcription assay directly supports the hypothesis that FET fusion protein condensates can recruit Pol II to activate gene transcription. From the methodological point of view, our transcription assay can be used as a complementary platform of in vivo or in vitro assays[7,27], to demonstrate the causality of the biomolecular condensates functions in general gene control[11]. For example, previous in vivo studies also suggested a potential correlation between coactivator condensates at super-enhancers and transcription activities[9,13].

Taken together, these insights help us understand how FET fusion oncoprotein condensates lead to transcription activities, which is one of the most important biological properties of biomolecular condensates[7,11], and also provide new tools and insights into cancer therapeutic development.

## Methods

**Construction of bacterial expression plasmids.** EWSR1, FLI1, and FUS genes came from human cDNAs. The genes coding Gal4 (1-147), MS2 coat protein, and Pol II CTD$_{N26}$ (POLR2A) were obtained from Addgene (Plasmid nos. 26264, 103831, and 35175). The vector for bacterial protein expression was pRSF-Duet (Novagen). We modify this vector to prepare a 7× His-tagged vector (His vector), a 7× His-GFP-tagged vector (GFP vector), a 7× His-mCherry-tagged vector (mCherry vector), and a 7× His-SNAP-tagged vector (SNAP vector). All 7× His

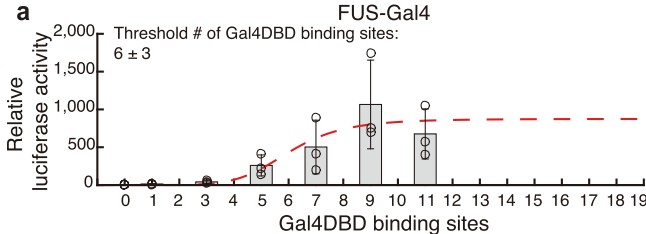

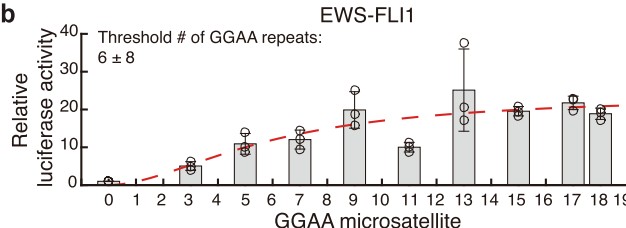

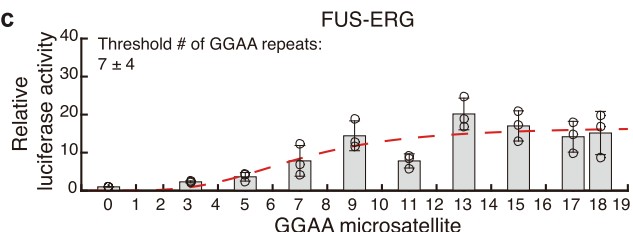

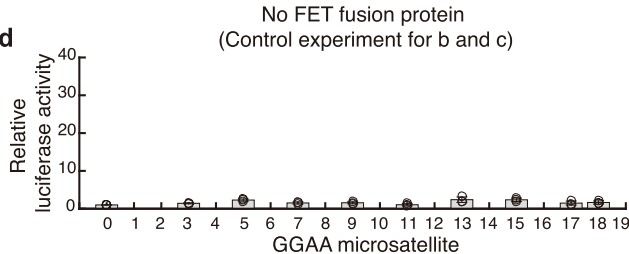

**Fig. 5 The number of GGAA microsatellites is highly associated with the gene transcriptional regulation by FET fusion proteins. a–d** Dual-luciferase assays were performed 24 h after transfection of 293 T cells with overexpression of either FUS-Gal4 a, EWS-FLI1 b, FUS-ERG c, or no fusion proteins d and a firefly luciferase vector possessing the indicated number of consecutive binding motifs preceding a minimal promoter. Data are normalized to relative luciferase activity of empty vector condition. Independent dual-luciferase assays were repeated three times ($n = 3$). Error bars, mean ± s.d. The red dashed line represents Hill function fitting. Error bars of the fitting parameters represent 95% confidence intervals obtained through Hill function fitting.

can be cleaved by TEV protease. FUS low complexity domain (LCD) (1–212) and Gal4 were fused with a 4× GGS linker, and then cloned to the vectors to prepare plasmids expressing differently tagged FUS-Gal4. Without the 4× GGS linker, EWSR1 (1–265) and FLI1 (220–453) were fused together to prepare the plasmid with the same EWS-FLI1 sequence found in Ewing sarcoma patients, and then cloned to the SNAP and GFP vectors. The first half of Pol II CTD were fused with mCherry tag with a 4× GGS linker, and then cloned to the His vector to prepare the plasmid of His-tagged Pol II CTD$_{N26}$-mCherry. T7 RNAP gene was a PCR product from E.coli BL21 (DE3) and fused with Pol II CTDN26 adjacent by a 4× GGS linker, then cloned to the His vector to prepare the plasmid of His-tagged Pol II CTD$_{N26}$-T7 RNAP. A 3× Flag tag was added behind the MS2 gene for Quantum dot labeling. All the sequences of the primers used for PCR were listed in Supplementary Table 1.

**Protein purification.** Plasmids of FET fusion proteins and MS2-3× Flag were transformed into E. coli strain BL21 (DE3), and then cultured overnight on LB agar plates. Monoclones were inoculated into 1l LB, and then were cultured until OD$_{600}$ reached 0.8. Bacteria were induced with 0.5 mM IPTG at 18 °C overnight. Cultures of FET fusion proteins were centrifuged at 4500 g. Cells were re-suspended in a

lysis buffer (50 mM Tris-HCl (pH 7.4), 1M KCl, 1M Urea, 10 mM Imidazole, 1.5 mM β-Mercaptoethanol (βME), and 5% Glycerol), and then sonicated. The supernatant was collected after centrifugation at 20,000 g for 30 min. Ni-NTA was used to bind these His-tagged proteins, and the beads were washed with a washing buffer (50 mM Tris-HCl (pH 7.4), 1M KCl, 1M Urea, 50 mM Imidazole, 1.5 mM βME, and 5% Glycerol). Finally, the protein was eluted by an elution buffer (50 mM Tris-HCl (pH 7.4), 1M KCl, 1M Urea, 500 mM Imidazole, 1.5 mM βME, and 5% Glycerol), and stored at −80 °C in the lysis buffer after further purification by gel filtration with a Superdex 200 column (GE Healthcare, USA). MS2-3× Flag was purified similarly except for the salt concentration which is 500 mM KCl and no Urea in the buffer. All the purified FET fusion proteins were loaded on a 10% SDS-PAGE gel to check the molecular weight and the purity (Supplementary Fig. 1a, b).

Purification of Pol II CTD$_{N26}$-T7 RNAP was quite same as FET fusion proteins, the Ni-NTA lysis buffer contains 50 mM Tris-HCl (pH 7.5), 100 mM NaCl, 5 mM βME, and 20% Glycerol. The Ni-NTA wash buffer contains 50 mM Tris-HCl (pH 7.5), 100 mM NaCl, 10 mM Imidazole, 5 mM βME, and 20% Glycerol. Finally, the proteins were eluted by elution buffer containing 500 mM Imidazole, and changed buffer into storage buffer (40 mM Tris-HCl (pH 7.5), 5 mM βME, 100 mM NaCl, 0.1 mM EDTA, and 50% glycerol)[44].

A plasmid containing Pol II CTD$_{N26}$-mCherry was transformed into E. coli strain Rosetta (DE3), and was purified similarly as FET fusion proteins, except the different lysis buffer (40 mM Tris-HCl (pH 7.4) and 500 mM NaCl), washing buffer (40 mM Tris-HCl (pH 7.4), 500 mM NaCl, and 40 mM Imidazole,), and elution buffer (40 mM Tris-HCl (pH 7.4), 500 mM NaCl, and 500 mM Imidazole). Protein was stored at −20 °C after purification in the storage buffer (40 mM Tris-HCl (pH. 7.4), 150 mM NaCl, and 50% glycerol).

The GFP molecule fused in all constructs in this work was the superfolder GFP (sfGFP), which contained a single mutation (A206K) and can avoid the weak dimerization of GFP[45].

**Electrophoretic mobility shift assay (EMSA).** EMSA experiments (Supplementary Fig. 1) were performed to test the binding affinity of proteins DNA. For Gal4 related proteins, DNA with/without a Gal4 target site (17-bp) was used (25-bp, 0.0625 μM). The Gal4DBD binding site was in the middle, connecting a random 4-bp sequence on either side. For FLI1 related proteins, DNA with GGAA microsatellites was used (306-bp, 0.1 μM), and scrambled DNA was amplified from Lambda DNA used for control experiment (306-bp, 0.1 μM). All DNA fragments were labeled with Quasar-670 at the 5' end. For MS2-3×Flag, RNA transcribed in vitro containing 6× MS binding sites was used. The working buffer included 40 mM Tris-HCl (pH 7.5), 150 mM KCl, 2 mM MgCl$_2$, 1 mM DTT, and 0.2 mg/ml BSA. The working buffer for EMSA was the same as the working buffer for DNA Curtain's experiments.

The samples with a different molar ratio of [protein]: [substrate] were pre-incubated for 30 min at room temperature, and then were loaded on a 8% native polyacrylamide gel electrophoresis (PAGE) gel. The size of the gel was 1.5 mm thick and 20 cm long. The gel was run in 1× TBE buffer (0.1 M Tris-base, 0.1 M Boric acid, and 2 mM EDTA) under 100 V voltage for 45 min. The protocol of 8% native PAGE gel (10 ml) was: (1) 2.67 ml Acrylamide/bis (30% 29:1; Amresco); (2) 2 ml 5× Tris/borate/EDTA (TBE) electrophoresis buffer; (3) 166 μl Ammonium persulfate (APS, 10%); (4) 8 μl TEMED (Amresco); (5) Replenish to 10 ml with distilled H$_2$O. The TBE PAGE gel for RNA substrate contained 4.5% Acrylamide/bis.

For FLI1 related proteins, the DNA substrates were incubated with indicated concentrations of proteins in the working buffer for 30 min under room temperature. 1.2% agarose gel was used. The gel was run in 1× TBE buffer under 120 V voltage for 30 min. The protocol of 1.2% agarose gel was dissolving 0.36 g agarose powder in 30 ml 1 × TBE buffer and cooling it down until it became solid gel. DNA substrates were labeled with Quasar670, and imaged by an Amersham Typhoon RGB (with a 635 nm laser and Cy5 670BP30 filter).

**Luciferase assay.** The synthesized fragments containing a different number of binding motifs were cloned upstream of the minimal promoter element of the pGL4.27 firefly luciferase vector (Promega, Cat. E8451). The 293 T cells were transfected with an experimental reporter plasmid, the Renilla luciferase plasmid (Promega, Cat. E6921) with EWS-FLI1 cDNA, FUS-ERG cDNA, or FUS-Gal4 cDNA. Dual-luciferase assays including firefly luciferase and Renilla luciferase were performed after ~24 h of transfection by following the instruction of luciferase assay system of Promega (Cat. E1910). For each experiment, the fluorescence intensity of Firefly was normalized by the Renilla luciferase intensity and compared with the normalized intensity of empty vector condition. Each experimental condition was performed three times. Hill equation was used to fit the result (red dash line in Fig. 5a–d), and the threshold number of binding motifs was fitted. Error bars represent 95% confidence intervals obtained through Hill function fitting. The Hill equation was:

$$\text{Relative luciferase activity} = \frac{A(\text{motif number})^n}{(\text{motif number})^n + (\text{Threshold motif number})^n}$$

(1)

Where $A$, $n$ (Hill coefficient), and Threshold motif number were three fitting parameter.

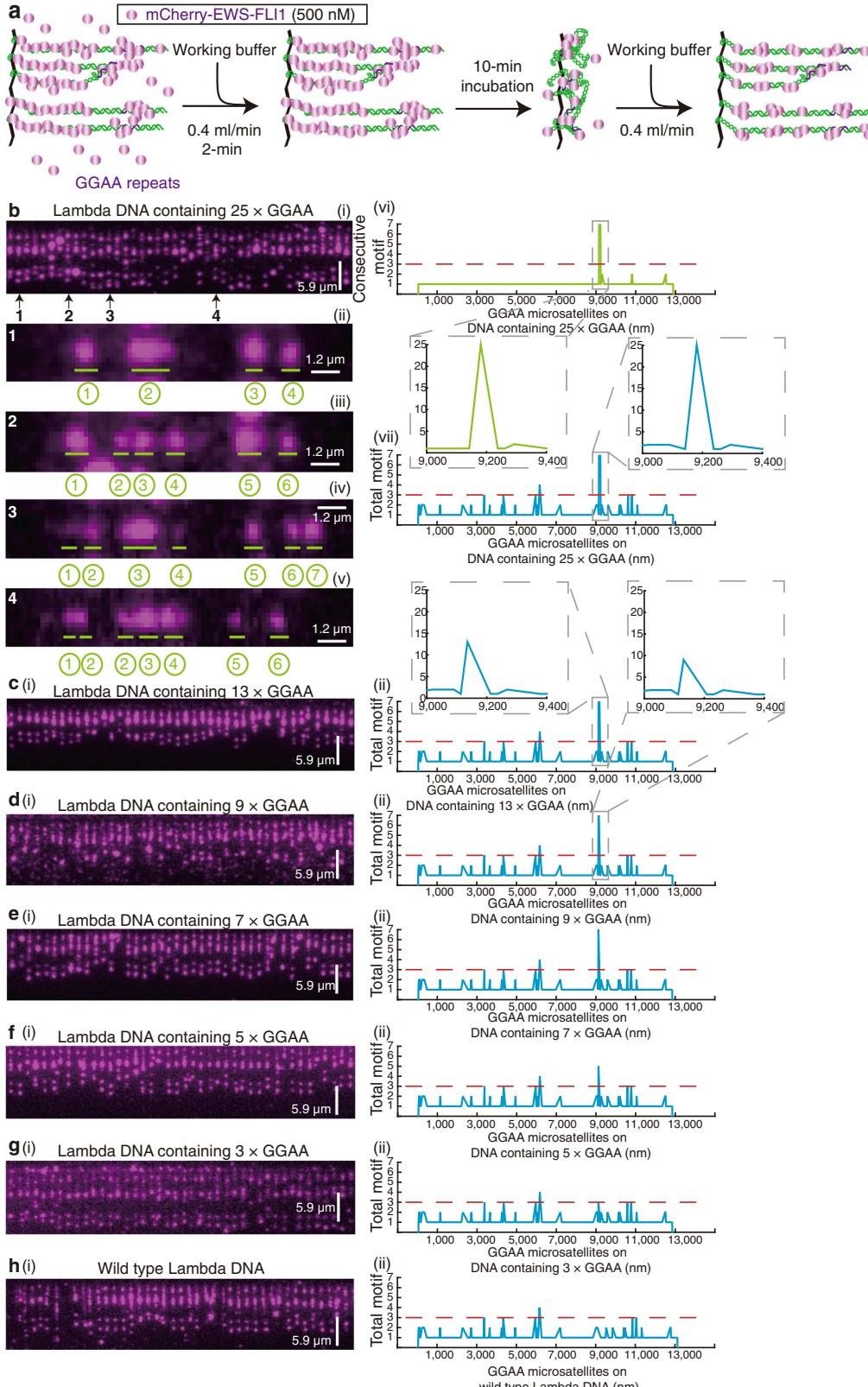

### In vitro droplet experiment and data analysis

*Samples preparation.* Three different kinds of biological samples were prepared: (i) Dilute the fusion proteins to 150 mM KCl (0.15× storage buffer), preparing the indicated concentrations (Fig. 1 and Supplementary Fig. 10); (ii) The DNA samples were named as 25× GGAA repeats, 0×, 5×, 7×, 9×, and 11× Gal4DBD binding sites, representing the number of 'GGAA' or 'CGG AGG ACA GTC CTC CG'. GFP labeled fusion proteins were mixed with/without these DNA samples at room temperature (Fig. 1 and Supplementary Fig. 10); (iii) For the Pol II CTD$_{N26}$-mCherry recruitment experiment: GFP-FUS-Gal4, GFP alone, and GFP-Gal4DBD were diluted to 150 mM KCl, and Pol II CTD$_{N26}$-mCherry was then added into the sample to satisfy the final concentrations as indicated in Fig. 3c–g.

*Experiment setup.* All in vitro samples were dripped into the 384-well plates (Cellvis). The mixed samples with different molar ratios and a different number of

**Fig. 6 DNA binding motifs beyond a threshold number drive the formation of FET fusion protein condensates. a** Strategy for detecting EWS-FLI1 molecules forming biomolecular condensates on DNA. **b**(i) A representative wide-field TIRFM image of DNA Curtains was taken 10 min after incubation with 500 nM of mCherry-EWS-FLI1 (top panel). (ii–v) Additional representative wide-field TIRFM images taken at a higher magnification showing the DNA substrate and EWS-FLI1 recruitment. DNA substrates used were Lambda DNA containing 25× GGAA repeats. (vi, vii) Line plot showing the hypothetical distribution of the number of puncta that would be visible if EWS-FLI1 recruitment was driven by a threshold number of consecutive motifs (vi) or total motifs (vii). Puncta are predicted when the number of motifs is equal to or above the threshold number of GGAA motifs as indicated by a dashed red line in respective line plots. Both inserts were the consecutive motif or total motif analysis near the region containing 25× GGAA repeats. **c–h** Lambda DNA containing 13× GGAA repeats c, 9× GGAA repeats d, 7× GGAA repeats e, 5× GGAA repeats f, 3× GGAA repeats g, and wild-type Lambda DNA h. (i) Same as **b**(i); (ii) Same as **b**(vii).

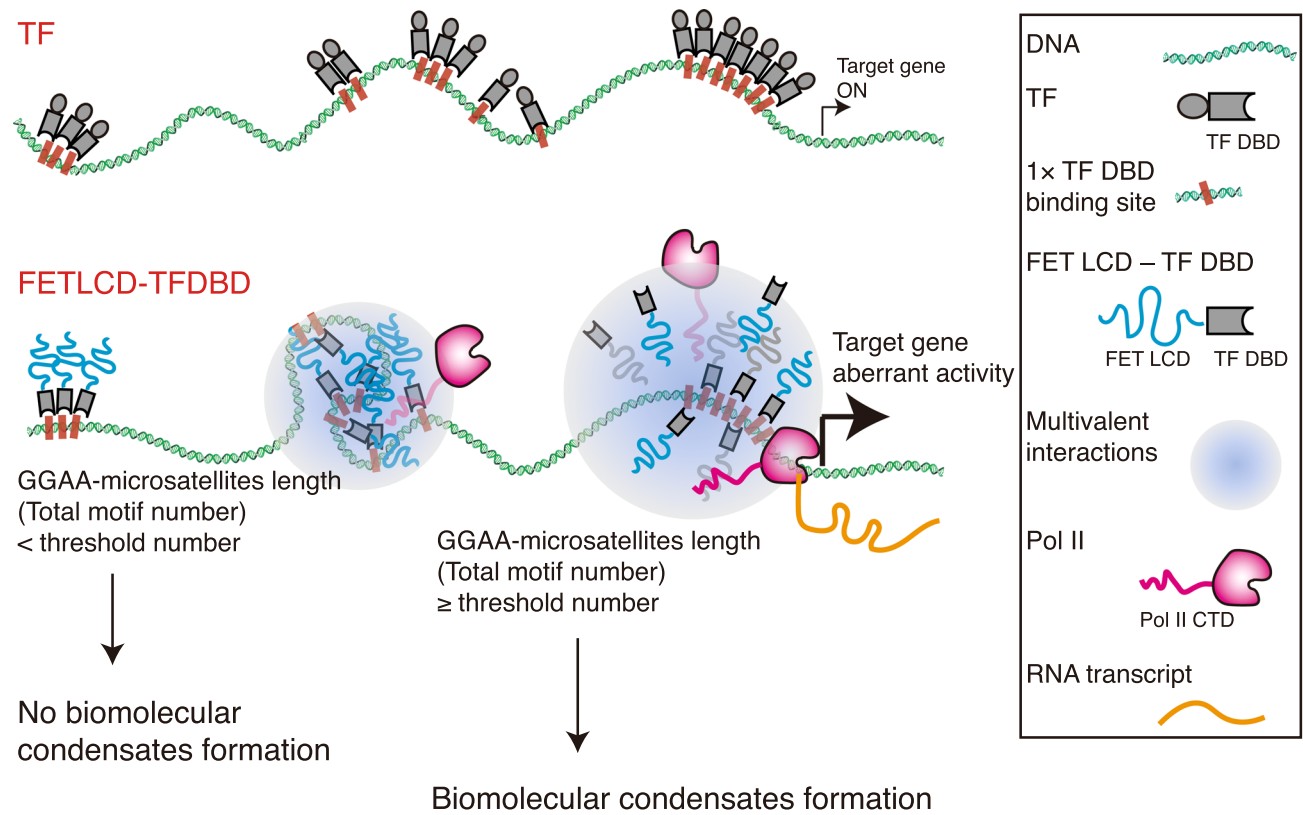

**Fig. 7 The model.** A schematic concept of a biophysical mechanism involving loci-specific biomolecular condensates of FET fusion oncoproteins promoting aberrant gene transcription.

Gal4DBD binding sites were visualized by a Nikon inverted microscope (Ti-Eclipse) with a 90× oil objective and a camera (HAMAMATSU ORCA-Flash4.0 V3 Digital CMOS camera C13440-20CU, Yihan Lin Lab at Peking University), and the output was a 2D image file (Supplementary Fig. 10a–c). The mixed samples with different fusion proteins, as well as the data acquisition of Pol II CTD$_{N26}$-mCherry recruitment and FRAP experiments were performed by a confocal laser scanning microscope (Nikon A1RMP, 100× oil objective, SLSTU-Nikon Biological Imaging Center at Tsinghua University). Laser power was adjusted to 100% for photo-bleaching. The photobleaching time for: (i) Fusion proteins was 1 s (488 nm laser); (ii) Pol II CTD$_{N26}$-mCherry recruitment was 16 s (561 nm laser and Fig. 3c–g). The first frame was immediately taken after bleaching, following by a chronological series of photos with a time interval of 10 or 20 s. For all FRAP experiments: (1) The time point for photobleaching was defined as 0 s; (2) The experiments were conducted at room temperature; (3) Error bars represent s.d.; (4) The normalized intensity came from Eq. 2 below.

　Normalization for FRAP data

$$I_{\text{Normalized}}(t) = \frac{\left(I_{\text{Bleached}}(t) - I_{\text{Background}}(t)\right) / \left(I_{\text{Bleached}}(t1) - I_{\text{Background}}(t1)\right)}{\left(I_{\text{Unbleached}}(t) - I_{\text{Background}}(t)\right) / \left(I_{\text{Unbleached}}(t1) - I_{\text{Background}}(t1)\right)}$$

(2)

*Image analysis (background subtraction).* In order to remove the background, we imported the image into the MATLAB software (R2015a, 64-bit, February 12, 2015) as a matrix with double precision, then the numbers in the matrix

corresponded to the brightness of the image. Centering at an arbitrary point in the matrix, we drew a square with the side length of 41 pixels and then compared the mean gray value inside the square to the brightness of the center. If the latter was significantly lighter than the former (in practice, a difference of 0.1 is considered significant), the center was treated as part of the foreground; otherwise, thrown into the background. By practicing the above algorithm on all the points in the image to determine the background and subtracting it to zero, we were able to get rid of overlie background. Using a sufficiently large structuring element (a disk with the radius of 50 pixels is used) to carry out a morphological opening on the initial image, we can estimate the background brightness of the image, which gave us more uniform foreground if subtracted. Then measured the area and intensity of the droplets in the processed images with ImageJ (Version: 2.0.0-rc-59/1.51k, http://imagej.net/Contributors): Analyze particles.

**Chromatin immunoprecipitation sequencing (ChIP-seq), RNA-seq, and bioinformatics analyses**
*Searching for GGAA repeat regions.* The human reference genome (hg19) was scanned for the GGAA motif sequence and the reverse complement sequence TTCC using Biostrings[46] and BSgenome[47] R packages. A custom script was utilized that identified regions with more than one GGAA-motifs not separated by more than 20 non-motif nucleotides. These regions therefore start and end with the motif sequence. Only repeat regions that occurred on separate strands were considered further. These repeat regions were annotated with their nearest gene using the ChIPseeker[48] R package.

*RNA-seq and ChIP-seq analysis*. The RNA-seq (NCBI SRA059239) and ChIP-seq (NCBI Omnibus GSE99959) datasets used in this work were previously published[38,49] and re-analyzed as previously described[50]. Briefly, for RNA-seq analysis, sequencing reads from RNA-seq were pseudoaligned to the GRCh37 (hg19) human genome following quality control of fastq files, and transcript abundances calculated in transcripts per million (TPM) using Kallisto (https://pachterlab.github.io/kallisto/about.html)[51]. Prior to alignment phred quality score of fastq files was inspected using the fastq package. Differential gene expression following *EWS-FLI1* knockdown was determined using the R package DESeq2[52] with the Benjamini–Hochberg corrected *q* value cutoff of 0.05. For ChIP-seq analysis, sequence reads were assessed for quality using FastQC (Version 0.11.8, https://www.bioinformatics.babraham.ac.uk/projects/fastqc/). Reads were trimmed to remove adapter sequences and trimmed to ensure quality using Trim Galore (Version 0.4.3, https://www.bioinformatics.babraham.ac.uk/projects/ trim_galore/). Reads were then aligned to the human reference genome (hg19) using Bowtie2 (Version 2.3.4, https://sourceforge.net/projects/bowtie-bio/files/bowtie2/2.3.4/)[53]. Duplicate reads were removed and aligned reads and filtered for mapping quality ≥20 using SAMtools (Version 1.3.1, http://www.htslib.org/download/)[54]. Peaks were identified using MACS2 (Version 3.4, https://pypi.org/project/MACS2/) an FDR cut-off of 1%[55]. To assess whether GGAA-repeat regions overlapped with EWS-FLI1 binding sites more than one would expect by chance, permutation tests were conducted. Overlap was defined as a motif region with ≥1 bp overlap. Briefly, we tested the number of overlaps between EWS-FLI1 binding sites and GGAA-repeat regions (with at least three consecutive repeats) to that observed in a random universe of repeat regions. This randomization approach maintained the internal structure of GGAA-repeats. EWS-FLI1 binding sites were paired with overlapping microsatellite regions. If ≥2 peaks overlapped a single microsatellite, the peak which was closest to the center of the microsatellite was chosen. All the R packages used for bioinformatics analysis could be downloaded from https://bioconductor.org/packages/3.12/bioc/.

**Bulk biochemical assay for in vitro transcription**. In vitro transcription was conducted in a solution (20 µl) containing 100 ng template dsDNA (2000 bp), 40 mM Tris-HCl (pH 8.0), 10 mM NaCl, 5 mM DTT, 8 mM MgCl$_2$, 2 mM spermidine, 5 mM NTPs, and T7 RNAP (TAKARA, Cat. 6140) at 42 °C for 2 h. Then the dsDNA template was degraded by RNase free DNase I (10 Units) at 37 °C for 30 min. RNA was precipitated by absolute ethanol and dissolved in DEPC ddH$_2$O. The purified RNA was loaded onto a 2% agarose gel (Supplementary Fig. 7a).

**GGAA microsatellites (consecutive motif or total motif) analysis for DNA substrates on DNA curtains**. The distribution of GGAA microsatellites for DNA substrates on DNA Curtains was mapped in Fig. 6b(vi, vii), c(ii), d(ii), e(ii), f(ii), g (ii), and h(ii). GGAA microsatellites are defined as: (i) Consecutive motif, the maximum distance between any two GGAA repeats is ≤1 bp. Endogenous GGAA microsatellites in living cells comply with the consecutive motif; (ii) Total motif, the maximum distance between any two GGAA repeats is ≤20 bp.

**DNA curtains**

*Experimental setup*. The nanofabrication techniques 'electron-beam lithography and thermal evaporation' were applied to produce many nanofabrication barriers on a quartz microscope slide (G. Finkenbeiner, Inc). A single-channel flow cell was formed by using a double-sided tape to glue this nanofabricated and fused silica slide with a thin coverslip. The surface of the flow cell was passivated by a lipid bilayer, including DOPC (100 mg/ml), PEG-2000 DOPE (10 mg/ml), and biotinylated DOPE (0.5 mg/ml).

Step 1. Wash a clean flow cell with 3 ml lipid buffer (10 mM Tris-HCl (pH 8.0) and 100 mM NaCl).

Step 2. Inject 1 ml liposomes (40 µl liposome storage solution plus 960 µl lipid buffer). The 1 ml liposome solution is injected by hand at the bench using a syringe in four sequential steps (250 µl per step) with an incubation time of 5 min between each injection.

Step 3. The flow cell is washed with another 3 ml lipid buffer, and then incubated at room temperature for 30 min. This procedure produces a lipid bilayer on the flow cell surface.

Step 4. The flow cell is then washed with 3 ml BSA buffer (40 mM Tris-HCl (pH 7.5), 2 mM MgCl$_2$, 1 mM DTT, and 0.2 mg/ml BSA).

Step 5. The lipid bilayer, which contains a small fraction of biotinylated lipids, is then rinsed with 800 µl streptavidin buffer (10 µl streptavidin stock (1 mg/ml) plus 790 µl BSA buffer). The streptavidin buffer is injected in two steps (400 µl per step), and allowed to incubate for 10 min between each injection. Note that the streptavidin stock (1 mg/ml) is prepared by dissolving 5 mg streptavidin (Invitrogen A00045) into 5 ml distilled water. This streptavidin stock solution can then be divided into small aliquots and stored at −20 °C.

Step 6. The flow cell is washed with an additional 3 ml BSA buffer to remove any free streptavidin.

Step 7. 1 ml dsDNA sample (15–20 pM) diluted in BSA buffer is then injected into the flow cell. The dsDNA sample is injected by hand in four steps (200 µl per step), and the incubation time between injections is 5 min. After this step, the flow cell is installed on the microscope stage and coupled to the sample delivery system.

Step 8. Working buffer for proteins with 0.5 nM YOYO1 is injected into flow cell at a flow rate of 0.03 ml/min for 10 min, and the buffer flow pushes the tethered dsDNA to the leading edges of the chromium barriers. After 10 min, turn on the flow rate to 0.4 ml/min to extend dsDNA and stain dsDNA with YOYO1. The working buffer used in DNA Curtains was: 40 mM Tris-HCl (pH 7.5), 150 mM KCl, 2 mM MgCl$_2$, 1 mM DTT, 0.2 mg/ml BSA, and 0.5 nM YOYO1. For protein loading, both EWS-FLI1 and FUS-ERG were diluted to the indicated concentrations by the working buffer in 100 µl, and loaded into a 50 µl sample loop[25]. The blank working buffer was used to send the protein sample in the sample loop to the chamber[25]. When proteins reached into the chamber, some of them interacted with and stayed on DNA, all other free proteins were washed out. We kept washing the chamber for 5 min.

*Injection of FET fusion proteins into the chamber*. For FUS-Gal4 (Fig. 3a(i–iv)), we injected around 2 ml working buffer including 150 nM FUS-Gal4 into the chamber for 5 min (totally ~1.8 × 10$^{14}$ FUS-Gal4 molecules) to form biomolecular condensates on DNA. However, when we repeated this method for EWS-FLI1, all DNA substrates stuck on the surface quickly. So we have to change the sample injection method. After many tests, we found that we can first inject 50 µl 500 nM EWS-FLI1 into a 50 µl extra loop, and use the blank working buffer to load the sample into the chamber (totally ~0.15 × 10$^{14}$ EWS-FLI1 molecules) (Fig. 2b)[25]. In this method, DNA substrates cannot stick on the surface.

*In vitro single-molecule transcription assay*. After FUS-Gal4 condensates formed on 7× Gal4DBD binding sites, the in vitro transcription mixture was prepared in a 100 µl solution, containing T7 transcription buffer (40 mM Tris-HCl (pH 8.0), 10 mM NaCl, 5 mM DTT, 8 mM MgCl$_2$, 2 mM spermidine), 5 µM mixture of ATPs, CTPs, and GTPs, 2.5 µM UTP-Fluor647 (GeneCopoia, Cat. C418B, for RNA transcripts labeling), and T7 RNAP (~0.4 unit) or CTD$_{N26}$-T7 RNAP. Then the mixture was loaded into a 50 µl loop, and injected into the flow cell by a flow rate of 0.4 ml/min. Once the mixture reached into the flow cell, the flow would be turned off right away and the whole reactions were incubated for 20 min at 37 °C, allowing sufficient transcription on DNA Curtains.

*Total internal reflection fluorescence microscope (TIRFM)*. All experimental data of DNA Curtains were acquired with a custom-built prism-type TIRFM (Nikon, Inverted Microscope Eclipse Ti-E), and the exposure time was 100 ms. The microscope was mounted with OBIS 488 nm, 561-nm LS 100-mW lasers. The real laser powers before the prism were measured: (i) 488 nm, 9.9 mW (20%); and (ii) 561 nm, 16.0 mW (20%), or 28.5 mW (50%).

*Lambda DNA cloning*. Wild-type Lambda DNA was purchased from NEB, and then 11×, 7×, 5×, and 1× Gal4 binding sites, as well as 25× microsatellite DNA, 13×, 9×, 7×, 5×, 3× GGAA sequence were inserted into the Lambda DNA XhoI/NheI sites, respectively. The ligation products were packaged into MaxPlax™ Lambda Packaging Extracts (Cat. no. MP5105) followed the instructions provided by Lucigen and let the phage plaques grow bigger on top agar plates. Afterward, a large amount of phage grew in LB broth with NZCYM *E. coli* cell were harvested through centrifugation, and Lambda DNA genome was purified from the supernatant. After getting Lambda DNA with the cloned binding sites, Biotin tag was added to the cos site of Lambda DNA by hybridization and T4 DNA ligase ligation.

*The DNA substrates preparation*. Biotinylated primer was annealed to Lambda DNA (N3011, NEB). The annealing protocol was: 100 µl (500 ng/µl) Lambda DNA was mixed with 1 µM biotinylated primer, and then the sample was incubated at 65 °C for 5 min, and the temperature was slowly decreased to the room temperature for another 45 min. Then 10× T4 DNA ligase buffer and 5 µl T4 DNA ligase (M0202, NEB) were added into the mix. The mix was incubated at 42 °C overnight. After overnight incubation, Buffer A (30% PEG 8000 and 10 mM MgCl$_2$) was added to dilute the mix. The volume of Buffer A was the half volume of the mix. The new mix was incubated at 4 °C with rotation for 1 day, and then centrifuged at 18,000 g for 5 min by a centrifuge (Lynx 4000, THERMO FISHER). Finally, the pellet including the biotinylated lambda DNA was dissolved by 100 µl TE150 buffer (10 mM Tris-HCl (pH 8.0), 1 mM EDTA (pH 8.0), and 150 mM NaCl).

*Single-tethered DNA curtains experiments*. One end of a DNA substrate was tethered to an individual lipid in the supported lipid bilayer within a microfluidic chamber by a biotin-streptavidin interaction. The working buffer flow can extend DNA substrates. When the flow is off, DNA molecules will shrink back to the barriers, and all binding events on DNA will also move with DNA. However, all signals of noise that are stuck on the single-molecule surface cannot move when the flow is off. This is an important benefit of DNA Curtains, which can distinguish these signals on DNA from the signals of noise on the single-molecule surface.

*Student's t test*. Statistical significance was evaluated based on Student's *t* test (Prism 7 for Mac OS X, Version 7.0c, March 1, 2017, GraphPad Software, Inc.). The *t* test was unpaired, and the *p* value is two-tailed. We assumed that both

populations has the same s.d. *p* value style: GP: 0.1234 (ns), 0.0332 (*), 0.0021 (**), 0.0002 (***), < 0.0001 (****). Confidence level: 95% (Definition of statistical significance: $p < 0.05$).

**Bootstrap analysis**. Matlab function 'bootci' (bootstrap confidence interval, R2015a; MathWorks, Inc., Natick, MA, USA) was used to calculate the error bars for binding position distributions (Fig. 3b(ii), f(v) and Supplementary Fig. 9). There are two parameters inside the function 'bootci': (1) bootstrap confidence interval, (100× (1-alpha)). When alpha = 0.3, bootstrap confidence interval = 70%, which was used in this paper; (2) repeat, which is the number of bootstrap samples used in the computation (nboot). Repeat = 1000 in this paper. For any normally distributed dataset, 68.27% of the values lie within one standard deviation of the mean, therefore, our choice of 70% confidence intervals for the bootstrapped data provides a close approximation to expectations for one standard deviation from the mean.

**Image tracking analysis of DNA end**. We used two algorithms to track DNA end: (i) We found the 'breakdown' point directly by fitting the brightness array to a step function, choosing the point with maximal drop as the end of the DNA chain; (ii) We found the first three points that was brighter than the average value of the DNA area from the bottom up, and took the median position of the points as the end of DNA chain. We also tried several ways to determine the light and dark threshold, including using the average value or a clustering algorithm (The 'fminbnd' function in the Matlab software). Finally, we integrated them into six methods that contributed six positions. After sorting the points by position, we calculated the variance of every three contiguous points, chose the three points with minimal variance, and took its average position as the final chosen position.

**Boxplot**. The function of "boxplot" in MATLAB software (R2015a, 64-bit, February 12, 2015) was used to plot the boxplots in Fig. 2l, Supplementary Figs. 3e, 5b–d, 8, and 10d. For each boxplot, the red bar represents the median. The bottom edge of the box represents 25th percentiles, and the top is 75th percentiles. Most extreme data points are covered by the whiskers except outliers. The '+' symbol is used to represent the outliers.

**Statistics and reproducibility**. The in vitro droplet assays in Fig. 1a, b, d, g, h, and j was repeated three times. Figure 1c came from 1a, 1e came from 1d, 1i came from 1g, and 1k came from 1j. The in vitro droplet assays in Fig. 3c, f, and g were repeated three times. Figure 3d came from 3c. The DNA Curtains experiments in Fig. 4a–e and g were repeated three times. The DNA Curtains experiments in Fig. 6b–h were repeated three times.

**Reporting summary**. Further information on research design is available in the Nature Research Reporting Summary linked to this article.

## Data availability

The data that support the findings of this study are available from the corresponding author upon reasonable request. Source data are provided with this paper.

## Code availability

Image analysis was performed using Open source image processing software ImageJ (Version: 2.0.0-rc-59/1.51k, http://imagej.net/Contributors). All the R packages used for bioinformatics analysis could be downloaded from https://bioconductor.org/packages/3.12/bioc/.

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

## Acknowledgements

We thank Dr. Eric Greene (Columbia University), Dr. Yihan Lin and Dr. Xiong Ji (Peking University), and members of the Z.Q. laboratory for comments on the manuscript. This work was supported by NSFC Grant No. 31670762 to Z.Q., 31871443 to P.L., the National Key R&D Program (2019YFA0508403) to P.L., and SULSA-PECRE/Royal Society IES\R2\192078 to X.H. M.M. is a recipient of Medical Research Council (MRC), UK DTP Ph.D. studentship.

## Author contributions

L.Z. prepared biological samples, designed and conducted all experiments, and performed data analysis. J.C. assisted L.Z. for luciferase assays. Z.G. assisted L.Z. for the protein purification. G.Z. and L.Z. conducted in vitro droplet experiments and data analysis. M.M. and X.H. conducted bioinformatics analysis for ChIP-seq data, and X.H. wrote the manuscript. L.W., Y.G., and R.L. assisted G.Z. P.L. supervised the project and wrote the manuscript. Z.Q. supervised the project, conducted theoretical calculations, experimental designs, and data analysis, and wrote the manuscript with input from all authors.

## Competing interests

The authors declare no competing interests.
