## [Peer Review File · Nature Communications]

REVIEWER COMMENTS

Reviewer #1 (Remarks to the Author):

Manuscript Identifier:

Title: Loci-specific phase separation of FET fusion oncoproteins promotes aberrant gene transcription

Zuo and colleagues investigate the important question of what the functional role of biomolecular condensates might be. In this report, the authors focus on the FET-family proteins EWS1 and FUS, both of which have been implicated in neurodegenerative diseases and cancers. Fusions with oncogenic proteins can lead these FET-family fusion proteins to drive sarcomas. To address how these fusion proteins might undergo liquid-liquid phase separation (LLPS) whilst engaging in transcription, the authors use DNA curtain single-molecule assay. First, they show that a model FUS-Gal4 fusion protein from a previous study as well as a patient-derived EWS-FLI1 fusion both undergo LLPS in vitro (Figure 1). These droplets display liquid-like properties, but DNA either has a negligible or minor effect on droplet fluidity (Figure 1). The authors then test the EWS1-FLI1 fusion protein with the DNA curtain assay, in which DNAs are arranged length-wise on a passivated single-molecule surface and proteins are flowed onto the surface so that they can interact with the DNA; a single-molecule resolution is attained with TIRF microscopy (Figure 2). The authors demonstrate that the EWS-FLI1 construct forms bright punctae, and the intensities of these punctae scale with protein concentration (Figure 2). The fusion protein FUS-Gal4 likewise can interact with Gal4 DNA response site, and a Pol II truncation can be recruited into in vitro droplets and these punctae on the single-molecule surface (Figure 3). By fusing the Pol II CTD to T7 RNAP, the authors further show that the FUS-Gal4 fusion protein promotes transcription of new RNA through the incorporation of fluorescent UTPs at these foci (Figure 4). Finally, the authors suggest that the number of DNA repeats must breach a critical threshold in order to promote transcription through an in vivo luciferase assay and analysis of a ChIP-Seq dataset (Figure 5). This length dependence does not appear to be present in their single-molecule assay (Figure 6).

This work addresses an important gap in knowledge in the LLPS field – what is the functional role of biomolecular condensates? The data are potentially interesting and the use of DNA curtain appears to be appropriate for some aspect. However, there are major deficiencies in the experimental design that need to be addressed with further controls and experiments. Throughout the paper, it is unclear whether there is actually a specific interaction between the fusion protein and the DNA sequences as the authors claim. This is especially evident because the addition of DNA has only minor or negligible effects on the droplet's liquid-like behavior in Figures 1D-K, the nonspecific binding widely evident in fluorescent images in Figures 2-4 and 6, and the inconsistency in the results between Figures 5 and 6. Although the interaction between the fusion protein and DNA appears to cause a transcription event as the authors claim, it seems unlikely that this is occurring at a specific loci and is rather a global event on their single-molecule surface. This manuscript cannot be published in its current state. The authors need to address the issues mentioned above which are more detailed in the following concerns.

Major Concerns

1. As mentioned above, it is unclear whether these fusion proteins bind the DNA constructs that the authors claim that they bind. Previous reports have suggested that FUS and FET family proteins do not bind double-stranded DNA (cite Schwartz/Cech paper), so presumably the DNA binding ability must come from the fused oncoprotein. The lack of apparent specificity makes it untenable to assert that fusion proteins form condensates at specific loci and also promote transcription at these specific loci.
 - a. In Figure S1, it is unclear how specific the DNA binding interaction between the protein and DNA is, considering the protein needs to be in great excess to have an apparent interaction with the DNA (in most gels, the unbound band does not decrease in intensity until the fusion protein is in 10+-fold excess compared to the DNA). Moreover, the apparent shift in the bound complex as the EWS-FLI1 protein concentration increases implies that there are conformational changes that are responding to protein concentration, something that would likely not occur if it were a stably bound monomer or

dimer complex. This instead implies that the DNA is trapped in an aggregate (or possibly condensate) formed by the fusion protein. A specific multivalent interaction would instead show laddering (cite Schwartz/Cech paper). The binding between FUS-Gal4 and the Gal4 sequence does not appear to be visible in Figures S1D/F/G, or is barely above background. Altogether this implies that there is some sort of nonspecific electrostatic interaction between the positively-charged FET protein RGG repeats and the negatively-charged DNA. To address this issue, the authors should either justify why these particular DNA sequences were used or repeat these and the following experiments with a DNA sequence that has a more specific interaction with the protein complex.

b. In Figures 1D-K, the addition of DNA has no or little effect on the FRAP curves. This implies that the DNA is not modulating the multivalent interaction between the proteins in the condensate. To show that the DNA is specifically recruited to the droplets, the authors could delete the purported DNA binding domains (Gal4 for the FUS fusion, FLI1 for the EWS1 fusion) and monitor whether DNA is included.

c. In Figures 2-4 and as the authors note, there is a high background fluorescence along the DNA strand where the repetitive DNA motif is not present. This readily visible as pink punctae near the top of the image in Figures 2Cii and 6B/E/G, widespread fluorescence in Figure 3Bi, LLPS punctae near the top of the image in Figure 2Fi. In many cases, there is no visible fluorescence difference between the specific binding sequence and the nonspecific binding position, especially for the EWS-FLI1 construct. It seems likely that the interaction between the fusion protein and the DNA motif it is supposed to recognize is weak enough to allow nonspecific interactions at the same time. If this is the case, then a more specific binding sequence for the fusion protein is needed, a more inert DNA sequence to limit background is needed, or the protein concentration needs to be reduced to limit nonspecific binding. It may also be possible to photobleach or otherwise monitor the fluorescence intensity of the specific versus nonspecific interactions to determine whether more fluorescent proteins are bound to certain sequences.

2. It would be ideal to show that the phase separation occurs without the GFP or mCherry tags (Figure 1), which can lead to increased solubility or dimerization. The untagged droplets should be morphologically similar to those formed by the tagged fusion proteins. It would also be helpful to show that the liquid-like properties are not affected by performing FRAP with fluorescently-labeled DNA (e.g. A408 or Cy5 label or some other label that does not undergo FRET with GFP or mCherry) droplets for the untagged and GFP-/mCherry-tagged fusion proteins. These experiments are necessary to show that GFP and mCherry tags do not affect the fusion proteins' LLPS properties.

3. There is limited evidence that the punctae in Figures 2-4 and 6 are actually liquid-like. The authors note that they sometimes observe fusion events, but these should be quantified and included in the main Figures alongside the data noting the existence of the punctae.

4. The authors also note that a higher number of GGAA repeats causes increased transcription in Figure 5. However, the *in vivo* luciferase experiments do not definitively show that the fusion protein is responsible for the increase in transcription of these longer GGAA repeats. Another factor within their cell line could easily be reading these repeats and promoting transcription above the same threshold repeat length (which appears to be 5 or 6 in Figures 5A-C). The authors need to perform a control without a fusion protein co-transfected to show that there is either a shift in the number of repeats needed for transcription or that the intensity of transcription is higher because of the fusion protein co-transfection. Moreover, the ChIP-Seq data from Figures 5D-G only weakly corroborates the notion that transcription increases with GGAA repeat length (especially Figures 5E/G; Figure 5F is not significant). The way to further stress the importance of the GGAA repeat length is to repeat the experiments from Figures 3-4 with the fusion proteins but vary the length of the GGAA repeat. If a difference in the apparent transcription scales with the GGAA repeat length, this would further corroborate the *in vivo* evidence shown in this figure and lend weight to the assertion that the fusion proteins – and not something else – are responsible for this increased transcription.

5. In Figure 3F, it is unclear whether this interaction between the Pol II FUS is actually specific or not. When the flow is turned off (Figure 7Fiii), the Pol II seems to preferentially localize to the top of the DNA strand away from the Gal4 DBD binding sites. As the authors show that the Pol II CTD can diffuse into and out of the condensates in Figure 3D-E, it seems like the Pol II CTD is diffusing out of the condensates in Figure 7Fii and preferentially interacting with another part of the DNA strand in Figure

7Fiii. It may be possible that the Pol II CTD is only localizing with the condensate when the flow is on because of the higher local viscosity of the condensates trapping the Pol II CTD rather than through a specific interaction. The low in vitro transcription observed in Figure S4J observed with physiological salts further supports the notion that this interaction is exceedingly weak. The authors should test for a specific interaction between FUS and the Pol II CTD and whether the binding between FUS, the Pol II CTD, and the DNA is cooperative.

Minor Concerns

6. The authors' title states that this is "aberrant" gene transcription – but we have no way to know whether this is aberrant or not given that there are no controls with either half of the fusion protein. Given that the vast majority of experiments use an artificial FUS-Gal4 system and a Gal4 DNA binding element, there is no way to conclude that this is an aberrant interaction because it is artificial.

7. It is unclear why Figure S2A and Figure S2C are included in the same figure as Figure S2B and Figure S2D. These should be separated.

8. The data from Video S2 is not quantified or shown in the main figures, and it should be included in a main or supplemental figure. Otherwise, it is difficult to evaluate the authors' claim that there is a significant increase in the green channel intensity upon addition of YOYO-1.

9. For the sake of consistency, the mCherry labeled images in Figures 3c-e should be colored magenta like other mCherry constructs.

10. The figures could be more clearly labeled to enable easier reading of the data. For instance, in many panels (Figures 1E, 1J, 3B, 3C, 3F, 4B, 6B, 6E, and 6G it is not clear what is being fluorescently labeled and what is being visualized simply by looking at the figure.

11. In general, more of the controls should be included in the main figures. Since the authors are using the DNA curtain assay and there is still a lot of room in many of these figures (especially Figure 4), it would make the paper more readable to include these controls and experiments in the main figures when appropriate. The figures are especially lacking in images of the control conditions in the main figures; for instance, Figure 4 should include sample images of all three lanes shown in the quantification (Figure 4C).

Reviewer #2 (Remarks to the Author):

In this article, the authors present a direct test of the causal links between condensates formed by fusion oncoproteins and increased transcription. This is an idea that has been proposed and discussed in the literature. Here, the authors deploy an in vitro, single molecule transcription assay, and show that the transcriptional output is directly tied to (i) condensate formation along the DNA; (ii) recruitment of RNA Pol II; (iii) and efficient and accurate generation of RNA transcripts that is governed, in part, by the valence of specific DNA motifs known as microsatellites.

The collection of data presented in this work are impressive, persuasive, and likely to garner significant interest within the readership of Nature Communications. Being able to demonstrate causal links, especially in vitro, will help place the field of biomolecular condensates on firmer footing. This MS goes a long way toward making this happen. It is well written, is quite comprehensive, and nothing is really left to chance. There are few points that would benefit from being addressed in a revised version. These are as follows:

1) The in vitro phase behavior of EWS-Fli1 and FUS-Gal4 raise an important question: These systems undergo phase separation at concentrations that are in line with the regimes established by Wang et al., 2018. However, instead of the RNA binding domains, we now have DNA binding domains. It would help if the authors were to discuss commonalities between the DBDs used here and the RBDs used in the molecular grammar work of Wang et al.

2) The methods and the figure captions do not provide information regarding the salt concentrations at which in vitro condensates are formed by the fusion proteins. One would have to presume that these condensates dissolve at monovalent salt concentrations of ca. 150 - 250 mM, and are stabilized at lower salt concentrations such as 75 mM NaCl. Is this correct? If so, might it be the case that the loss of transcriptional activity at 150 mM NaCl is due to the dissolution of condensates? An explanation of this issue would be very helpful.

3) The schematic and the overall mechanism that emerge could use some clarification. Do the authors propose that condensates form autonomously and dock onto DNA by displaying a multivalence of DBDs? Or do the condensates form around the DNA leveraging a combination of homotypic interactions (presumably involving the LCD and DBD) and heterotypic interactions (mainly involving the DBD and DNA but also the LCD and DNA) whereby the DNA weaves through condensates? The schematic in Fig. 7 appears to be consistent with both mechanisms and whether or not the data can adjudicate between these models is unclear. What would help is the interrogation of the phase behavior of either EWS-Fli1 or FUS-Gal4 in the presence of oligos comprising the GGAA motifs. This is a relatively straightforward experiment whereby one can titrate the concentration of GGAA motifs along one axis and the concentration of oncoprotein along the other axis and assess the contribution of motif interactions to condensate formation. If the phase boundary shows a strong synergy between the two concentrations, then there is an interplay of homotypic and heterotypic interactions and the model is a weave through rather than a dock onto DNA model.

Once these issues have been clarified, this MS should be acceptable for publication, at least from my perspective.

Reviewer #3 (Remarks to the Author):

In this manuscript, "Loci-specific phase separation of FET fusion oncoproteins promotes aberrant gene transcription," Zuo et al. purify recombinant FET fusion proteins and evaluate the ability of these fusion proteins to phase separate and recruit the RNA polymerase II C-terminal domain in vitro using a DNA curtains assay. They also develop a novel in vitro transcription assay based upon the DNA curtains technique to evaluate the mechanistic link between phase separation and local transcriptional activation. The main claims of the paper are that they have developed a novel in vitro single molecule assay for DNA-templated phase separation, that FET fusion proteins undergo phase separation at target loci, that phase separated condensates recruit RNA polymerase II and enhance transcription, and that they determined a threshold of fusion-binding DNA elements.

Zuo et al. have taken an innovative approach to address important questions regarding the contribution of phase separation to oncogenic FET-ETS fusion transcription factor function. It is widely speculated that phase separation at specific loci is a critical facet of FET-ETS function in pediatric tumors, but the evidence published to date is circumstantial. Direct evidence would represent a step forward for the field. This paper has several strengths conferred by using the DNA curtains assay, both as a stand-alone assay and as the basis for an in vitro transcription assay. The most striking finding of the paper is that recruited T7 polymerase initiates transcription even when recruited by a FUS condensate in the absence of a T7 promoter. The authors also spent a significant amount of time deeply considering the existing literature for EWS/FLI binding at GGAA microsatellites and worked to thoughtfully incorporate some of the more subtle pieces of the literature into the interpretation of their results. Unfortunately, this paper also had several key weaknesses that undercut the ability to meaningfully apply the data gathered here to EWS/FLI function. These include a lack of controls required to interpret the EWS/FLI experiments, EWS/FLI experiments that appear to not support the model put forward, and collection of data primarily from an artificial FUS-Gal4 construct. While the in vitro transcription finding is quite interesting and compelling, this manuscript cannot be published in its current form without addressing these weaknesses, outlined in the comments to authors below.

Major concerns:

EWS/FLI is a notoriously difficult protein to purify and work with. A Coomassie gel showing the purification of recombinant EWS/FLI and FUS-Gal4 used for experiments should be shown in the Supplement or Extended Data. This is required to show that the effects seen across all experiments are really coming from the purified protein and not an artifact of the inclusion of some impurity in the protein prep.

In a similar vein, the EMSA assays in Extended Data Figure 1 all suggest the authors have, indeed, purified functional protein, with the exception of panels f and g. These are lacking a shifted band at a higher molecular weight, and instead only show the loss of the lower band.

Figure 1 is currently uninterpretable without additional controls, including GFP alone. An appropriate negative control showing no phase separation with the tag is important, as well as the GFP-DBD (i.e. GFP-Gal4 and GFP-FLI) construct to show that the intrinsically disordered N-terminus of FUS and EWS is required for phase separation as predicted by the overarching model. Moreover, it's important to know whether or not DNA phase separates into droplets because of the binding affinity of the DBD in the droplet, or simply because of physical factors. Appropriate controls for this include a DNA-binding mutant (such as the R2L2 mutant of FLI) or DNA lacking the binding motif (i.e. GGAT microsatellites or scrambled DNA).

Figure 2 is currently uninterpretable without additional controls. These controls are very similar to those discussed above for figure 1 and include an mCherry alone, an mCherry-FLI DBD construct (data is included, but with no DNA curtains figure), an mCherry-EWS construct, and a DNA-binding mutant of EWS/FLI (i.e. R2L2). These constructs all need both a DNA curtains figure and the puncta intensity included in the plot. Alterations to the DNA sequence (i.e. GGAT repeats) would also be helpful.

Figure 2 does not show a single puncta at the microsatellite and instead shows EWS/FLI coating the DNA strand. The puncta at the microsatellite do not appear to be more intense than the puncta elsewhere, somewhat negating the discussion of the behavior focusing on microsatellite (page 6, second paragraph).

Does the YOYO-1 after EWS/FLI infusion match the EWS/FLI puncta throughout the DNA strand?

For Figures 3 and 4 (and supporting Extended Data), the authors use a FUS-Gal4 construct similar to that used by the McKnight lab. As for figure 1, negative controls with just the GFP and GFP-Gal4 need to be included in the phase separation assay with Pol II CTD.

Figures 3 and 4 show some interesting phase separation activity, Pol II CTD recruitment, and in vitro transcription. However, these data are not generally applicable to EWS/FLI as the authors have written. First, there are distinct differences in the number and density of critical (G/S)Y(G/S) repeats in FUS and EWS (and TAF15) that may have important biological relevance. Second, Gal4 and ETS transcription factors bind DNA in completely different manners. Gal4 homodimerizes such that 2 Gal4 proteins bind a single Gal4 motif, while ETS family member bind monomerically. Gangwal et al. (2010 Genes Cancer) and Johnson et al. (2017 PNAS) show that a single EWS/FLI binds monomerically to a 2X GGAA repeat. Moreover, Johnson et al. has recently suggested that the EWS domain of EWS/FLI alters the DNA-binding function of FLI to confer the ability to bind GGAA microsatellites, such that binding multiple EWS/FLI molecules at a microsatellite is mechanistically different than simply stringing together multiple high affinity sites, as is done here for Gal4. Unless the appropriate controls are used to evaluate the difference between multiple high affinity ETS sites and GGAA microsatellites with EWS/FLI the authors should revise portions of their paper where they apply finding from FUS-Gal4 to EWS/FLI. A more appropriate comparison might be EWS/ATF1 or EWS/ATF7 where the DBD of the fusion is also from a transcription factor which dimerizes to bind DNA.

Supplementary video 5 does not support the conclusion that EWS/FLI functions like FUS-Gal4. Fewer

Pol II CTD puncta are observed and they tend to be much lower on the DNA strand than a majority of the areas of high YOYO-1 intensity suggesting EWS/FLI is either not phase separating at the GGAA microsatellite and that Pol II CTD is not recruited to the non-microsatellite DNA where EWS/FLI is phase separating.

For the FUS-Gal4 experiments where phase separation is said to be mechanistically important (i.e. Pol II CTD recruitment and transcription initiation) the authors should directly test that by mutating the (G/S)Y(G/S) tyrosine residues thought to be critical for phase separation in FET proteins. This is actually a nice assay to do this, because unlike for EWS/FLI, disrupting the phase separation phenotype shouldn't impact the DNA binding activity of the construct.

While the luciferase assays in Figure 5 show similar activity thresholds as those previously identified by Gangwal et al and Johnson et al, similar concerns are present regarding the ability to compare FUS-Gal4 to EWS/FLI in luciferase assays. Just looking at the scale of the induced luminescence suggests the difference in DNA binding mechanism of ETS factors and Gal4 may play a role in assay function.

Figures 5d and 5e are basically the same analyses as was performed by Johnson et al. 2017 (PLoS ONE). The way the paper is currently written suggests that the identification of a threshold is a new contribution to the field, but this idea of a threshold has been previously shown by Gangwal et al (2008 and 2010) and Johnson et al (2017 PNAS and PLoS ONE). The authors add the analysis with the total motif and this particular addition should be highlighted as further supporting previous work, rather than as a novel finding.

For Figure 6, please include the data for the 3X, 7X, 9X, and 13X GGAA repeats. This is needed to draw the conclusion that "a threshold number of GGAA repeats is required" as stated in the text.

Figure 6 is impacted by the same issues as Figure 2.

Specifically that mCherry-EWS/FLI binds at many other regions than just the GGAA microsatellite. This undermines the purpose of the figure and suggest that consecutive GGAA repeats do not contribute to EWS/FLI binding at DNA. The authors attempt to explain this by showing that there is a threshold of the total number of GGAA repeats in a stretch of DNA rather than the consecutive motifs. However, in their analysis of the lambda DNA in Figure 6, they switch the definition from motifs within 20-bp (as previously published and from the analysis for Figure 5) to motifs within 160-bp. This is a much larger window and it is unclear that this holds up in a genome-wide analysis. They should keep the window for total motifs consistent between Figure 5 and Figure 6.

Together, the explanation that it is just simply the total number of GGAA motifs that is important for phase separation and Pol II recruitment, but not consecutive motifs, is contrary to much of the evidence in the field to date. It may be true in vitro, but the in vivo evidence suggests that consecutive motifs carry real biological significance. Consecutive motifs are more highly bound by EWS/FLI (ChIP-seq) and the change of a single base pair to generate a new string of consecutive motifs (Grünwald et al. Nat Genet 2016). Consider revising discussion of total vs. consecutive motifs with this in mind.

In the first paragraph of the discussion, the authors state that the "transcription assay confirms that recruited Pol II molecules indeed activate gene transcription, and hence possibly exerts aberrant transactivation in cancer patient cells." This is a bit of an overstatement, as the in vitro assay used an artificial construct for both the DNA-binder and Pol II. Additionally, the Supplemental Video 5 doesn't show robust Pol II recruitment to EWS/FLI-bound loci.

Minor concerns:

The first sentence refers to "lipid membranes-separated," in this phrase, the membrane should not be

plural. Find and replace instances of this to "lipid membrane-separated" throughout the paper.

The current nomenclature is "Ewing sarcoma," not "Ewing's sarcoma." Be sure to consistently use the correct nomenclature throughout the paper.

For Figure 1k, where are the pre-bleach intensities for the DNA-containing samples?

Page 6, paragraph 1: the second sentence needs to be re-written and likely split into two sentences to better communicate the rationale for using 25x GGAA microsatellites.

YOYO-1 staining needs to be introduced for readers unfamiliar with it before the discussion of Figure 2.

On page 13 the second to last line reads "to record the distribution of GGAA microsatellites in the A673 genome." This suggests to the reader that the authors have reanalyzed the ChIP-seq data with a custom A673 genome, not hg19 as discussed in the methods. This should just say "to record the distribution of GGAA microsatellites in the genome."

For Extended Data Figure 5, the y-axes should all be set equally.

On page 17, in the second paragraph the authors state that "not only LCD-LCD interactions, but also LCD-DBD interactions" are important has already been shown by Johnson et al (2017 PNAS). This reference should be added.

The file for Supplementary Video 3 is broken. I tried downloading it in the zip for reviewer and the source data, but it never worked.

**Decision on manuscript NCOMMS-20-21264-T.
July 8, 2020**

Dear Dr Qi,

Your manuscript entitled "Loci-specific phase separation of FET fusion oncoproteins promotes aberrant gene transcription" has now been seen by 3 reviewers, whose comments are appended below. You will see from their comments copied below that while they find your work of considerable potential interest, they have raised quite substantial concerns that must be addressed. In light of these comments, we cannot accept the manuscript for publication, but would be interested in considering a revised version that addresses these serious concerns.

We hope you will find the reviewers' comments useful as you decide how to proceed. Should further experimental data or analysis allow you to address these criticisms, we would be happy to look at a substantially revised manuscript. However, please bear in mind that we will be reluctant to approach the reviewers again in the absence of major revisions. We consider it particularly important that the conclusions be supported by additional experimentations that should include specific interaction between DNA and the protein and all the necessary controls.

We would normally ask to see a revised version of this paper within 6 months but we appreciate revisions may take longer than usual and can extend this timeline if the Covid-19 pandemic prevents you from undertaking any further work for a longer period - please do get back to us on this nearer the time.

We are committed to providing a fair and constructive peer-review process. Do not hesitate to contact us if you wish to discuss the revision or if there are specific requests from the reviewers that you believe are technically impossible or unlikely to yield a meaningful outcome.

When resubmitting, you must provide a point-by-point response to the reviewers' comments. Please show all changes in the manuscript text file with track changes or colour highlighting. If you are unable to address specific reviewer requests or find any points invalid, please explain why in the point-by-point response.

In addition to the above, you must comply with the following editorial requests; failure to do so will cause delays upon resubmission.

Best regards,

Minju Ha, PhD

Associate Editor

REVIEWER COMMENTS

Reviewer #1 (Remarks to the Author):

Manuscript Identifier:

Title: Loci-specific phase separation of FET fusion oncoproteins promotes aberrant gene transcription

Zuo and colleagues investigate the important question of what the functional role of biomolecular condensates might be. In this report, the authors focus on the FET-family proteins EWS1 and FUS, both of which have been implicated in neurodegenerative diseases and cancers. Fusions with oncogenic proteins can lead these FET-family fusion proteins to drive sarcomas. To address how these fusion proteins might undergo liquid-liquid phase separation (LLPS) whilst engaging in transcription, the authors use DNA curtain single-molecule assay. First, they show that a model FUS-Gal4 fusion protein from a previous study as well as a patient-derived EWS-FLI1 fusion both undergo LLPS in vitro (Figure 1). These droplets display liquid-like properties, but DNA either has a negligible or minor effect on droplet fluidity (Figure 1). The authors then test the EWS1-FLI1 fusion protein with the DNA curtain assay, in which DNAs are arranged length-wise on a passivated single-molecule surface and proteins are flowed onto the surface so that they can interact with the DNA; a single-molecule resolution is attained with TIRF microscopy (Figure 2). The authors demonstrate that the EWS-FLI1 construct forms bright punctae, and the intensities of these punctae scale with protein concentration (Figure 2). The fusion protein FUS-Gal4 likewise can interact with Gal4 DNA response site, and a Pol II truncation can be recruited into in vitro droplets and these punctae on the single-molecule surface (Figure 3). By fusing the Pol II CTD to T7 RNAP, the authors further show that the FUS-Gal4 fusion protein

promotes transcription of new RNA through the incorporation of fluorescent UTPs at these foci (Figure 4). Finally, the authors suggest that the number of DNA repeats must breach a critical threshold in order to promote transcription through an in vivo luciferase assay and analysis of a ChIP-Seq dataset (Figure 5). This length dependence does not appear to be present in their single-molecule assay (Figure 6).

This work addresses an important gap in knowledge in the LLPS field – what is the functional role of biomolecular condensates? The data are potentially interesting and the use of DNA curtain appears to be appropriate for some aspect. However, there are major deficiencies in the experimental design that need to be addressed with further controls and experiments. Throughout the paper, it is unclear whether there is actually a specific interaction between the fusion protein and the DNA sequences as the authors claim. This is especially evident because the addition of DNA has only minor or negligible effects on the droplet's liquid-like behavior in Figures 1D-K, the nonspecific binding widely evident in fluorescent images in Figures 2-4 and 6, and the inconsistency in the results between Figures 5 and 6. Although the interaction between the fusion protein and DNA appears to cause a transcription event as the authors claim, it seems unlikely that this is occurring at a specific loci and is rather a global event on their single-molecule surface. This manuscript cannot be published in its current state. The authors need to address the issues mentioned above which are more detailed in the following concerns.

We thank the reviewer for these important comments and questions, and conducted all new control experiments (see below), especially to prove these FET fusion proteins can bind to DNA containing their specific binding sequences. We have revised the relevant text, prepared new Supplementary Figures, and cited all relevant references

as suggested in the new manuscript. All changes in the manuscript text file were shown with red color highlighting.

Major Concerns

1. As mentioned above, it is unclear whether these fusion proteins bind the DNA constructs that the authors claim that they bind. Previous reports have suggested that FUS and FET family proteins do not bind double-stranded DNA (cite Schwartz/Cech paper), so presumably the DNA binding ability must come from the fused oncoprotein. The lack of apparent specificity makes it untenable to assert that fusion proteins form condensates at specific loci and also promote transcription at these specific loci.

a. In Figure S1, it is unclear how specific the DNA binding interaction between the protein and DNA is, considering the protein needs to be in great excess to have an apparent interaction with the DNA (in most gels, the unbound band does not decrease in intensity until the fusion protein is in 10+-fold excess compared to the DNA). Moreover, the apparent shift in the bound complex as the EWS-FLI1 protein concentration increases implies that there are conformational changes that are responding to protein concentration, something that would likely not occur if it were a stably bound monomer or dimer complex. This instead implies that the DNA is trapped in an aggregate (or possibly condensate) formed by the fusion protein. A specific multivalent interaction would instead show laddering (cite Schwartz/Cech paper). The binding between FUS-Gal4 and the Gal4 sequence does not appear to be visible in Figures S1D/F/G, or is barely above background. Altogether this implies that there is some sort of nonspecific electrostatic interaction between the positively-charged FET protein RGG repeats and the negatively-charged DNA. To address this issue, the authors should either justify why these particular DNA sequences were used or repeat

these and the following experiments with a DNA sequence that has a more specific interaction with the protein complex.

We thank the reviewer for these comments and questions. As suggested by the reviewer, we repeated all EMSA experiments in Supplementary Fig. 2. In order to achieve better imaging results, DNA substrates were labeled with Quasar670, and imaged by Amersham Typhoon RGB (with a 635 nm laser and Cy5 670BP30 filter). This method can effectively reduce the noise signals from GFP or mCherry tag in EMSA.

For EWS-FLI1, the 306-bp microsatellite DNA in this work was considered as the specific binding sequence in the previous references ¹⁻⁴. The 306-bp microsatellite DNA includes 25× GGAA repeats, which is a crucial microsatellite for EWS-FLI1 in Ewing sarcoma located within the promoter region of *NR0B1* ⁵. In new EMSA experiments, we designed a scrambled 306-bp sequence as a negative control DNA. Our new data show: (i) mCherry-EWS-FLI1 (Supplementary Fig. 2a), GFP-EWS-FLI1 (Supplementary Fig. 2b), and mCherry-FLI1DBD (Supplementary Fig. 2c) present significantly stronger binding to the microsatellite DNA compared to the scrambled DNA, suggesting the more specific interaction between the EWS-FLI1 and the DNA fragment containing 25× GGAA repeats; (ii) No DNA-protein complex was observed for GFP-EWSLCD (Supplementary Fig. 2d) and the microsatellite DNA, confirming that there is no interaction between EWSLCD and DNA fragment tested.

As for FUS-Gal4, in contrast to the 1× Gal4DBD sequence (5' - ATAT - CGG AGG ACA GTC CTC CG - AATA - 3'), we also designed a scrambled sequence (5' - ATAT - ATT TTT TAC AAT AGA AT - AATA - 3') as a negative control. Similarly the new results show: (i) mCherry-FUS-Gal4 (Supplementary Fig. 2e), GFP-FUS-Gal4 (Supplementary Fig. 2f), FUS-Gal4 (Supplementary Fig. 2g), and Gal4DBD

(Supplementary Fig. 2h) have much stronger binding to the 1× Gal4DBD sequence compared to the control sequence, suggesting a more specific interaction between FUS-Gal4 and Gal4DBD. (ii) GFP-FUSLCD (Supplementary Fig. 2i) does not bind to either the 1× Gal4DBD sequence or the control sequence, confirming that there is no interaction between FUSLCD and DNA fragment tested.

Additionally, we would like to further clarify the description of EWS-FLI1 and FUS-Gal4 used in this study. EWS-FLI1 and FUS-Gal4 are the fusions between the low complexity domain (LCD) of EWS or FUS and the DNA-binding domains (DBD) of transcription factor (TF) FLI1 or Gal4, respectively: (i) For EWS-FLI1 ⁶, the whole N-terminal SYGQ rich domain (1-264) of EWSR1 ⁷ was fused with the DBD (1-235) of FLI1. Here we noted that no RNA recognition motif (RRM) or RGG-rich regions (RGG) of EWSR1 were included in this EWS-FLI1 fusion. Our new EMSA result also confirmed that EWSLCD alone cannot bind to the 306-bp microsatellite DNA containing 25× GGAA repeats (Supplementary Fig. 2d); (ii) For FUS-Gal4, the N-terminal SYGQ rich domain of FUS (1-165) and also a small part (165-212) of the first RGG domain (165-267) ⁷ were fused with the DBD (1-147) of Gal4. Despite of the FUS-Gal4 molecule containing a small region of RGG, the new EMSA result confirmed that FUSLCD alone cannot bind to the 1× Gal4DBD sequence nor the control sequence (Supplementary Fig. 2i), suggesting there is no specific electrostatic interaction present between the positively-charged FET protein RGG repeats and the negatively-charged DNA for EWS-FLI1 and FUS-Gal4.

Taken together, our data demonstrated that EWS-FLI1 and FUS-Gal4 specifically bind to their known target sequences used in this study. We revised the relevant main text at Page 5 of the new manuscript, and also cited the relevant references suggested by the reviewer.

b. In Figures 1D-K, the addition of DNA has no or little effect on the FRAP curves. This implies that the DNA is not modulating the multivalent interaction between the proteins in the condensate. To show that the DNA is specifically recruited to the droplets, the authors could delete the purported DNA binding domains (Gal4 for the FUS fusion, FLI1 for the EWS1 fusion) and monitor whether DNA is included.

We thank the reviewer for these comments and questions. Following the reviewer's suggestion, we conducted the *in vitro* droplet experiments for FUS LCD and EWS LCD with loss of their DNA binding domains. For visualization, both proteins were labeled with a GFP tag.

Previous references^{8,9} showed that FET LCDs can form droplets at much higher concentrations. For example, Fawzi and coworkers found that at 200 μM FUSLCD can form droplets under certain experimental conditions, such as low temperature and high salt *in vitro*⁸. We repeated this experiment and confirmed that GFP-FUSLCD (Response Fig. 1.1a(i)) or GFP-EWSLCD (Response Fig. 1.1c(i)) cannot form droplets at a protein concentration up to 30 μM *in vitro*.

Next, we followed the previous references^{10,11} to add the molecular crowding agents (5% PEG8000). For FUSLCD, we observed that even at 10 μM concentration, GFP-FUSLCD can form droplets after adding 5% PEG8000 (Response Fig. 1.1a(ii)). However under this condition, the DNA substrates containing 11 \times Gal4DBD binding sites cannot be recruited to the droplets (Response Fig. 1.1b). Similarly for EWSLCD, at 12 μM concentration, GFP-EWSLCD can form droplets after adding 5% PEG8000 (Response Fig. 1.1c(ii)). When we added DNA substrates containing 25 \times GGAA repeats, the number of droplets dramatically decreased (Response Fig. 1.1d(i)). Though in this case the DNA can be recruited to the droplets to certain degree

(Response Fig. 1.1d(iii)), this might reflect the difference between EWSLCD and FUSLCD. Nevertheless, our EMSA data (Supplementary Fig. 2d) did not show any interaction at all between EWSLCD and DNA containing 25× GGAA repeats.

Response Fig. 1.1 | FET LCD cannot recruit DNA into the FET LCD condensates.

(a) 30 μM GFP-FUSLCD (No PEG8000) (i); 10 μM GFP-FUSLCD (5% PEG8000) (ii).

(b) 10 μM GFP-FUSLCD (5% PEG8000) mixed with 0.36 μM 326-bp dsDNA. (i) GFP channel; (ii) Quasar670 channel; (iii) Merge. DNA substrates contained 11× Gal4DBD binding sites and labeled with Quasar 670.

(c) 30 μM GFP-EWSLCD (No PEG8000) (i); 12 μM GFP-EWSLCD (5% PEG8000) (ii).

(d) 12 μM GFP-EWSLCD (5% PEG8000) mixed with 0.45 μM 306-bp dsDNA. (i) GFP channel; (ii) Quasar670 channel; (iii) Merge. DNA substrates contained 25× GGAA repeats and labeled with Quasar 670.

Merge. DNA substrates contained 25× GGAA repeats and labeled with Quasar 670.

c. In Figures 2-4 and as the authors note, there is a high background fluorescence along the DNA strand where the repetitive DNA motif is not present. This readily visible as pink punctae near the top of the image in Figures 2Cii and 6B/E/G, widespread

fluorescence in Figure 3Bi, LLPS punctae near the top of the image in Figure 2Fi. In many cases, there is no visible fluorescence difference between the specific binding sequence and the nonspecific binding position, especially for the EWS-FLI1 construct. It seems likely that the interaction between the fusion protein and the DNA motif it is supposed to recognize is weak enough to allow nonspecific interactions at the same time. If this is the case, then a more specific binding sequence for the fusion protein is needed, a more inert DNA sequence to limit background is needed, or the protein concentration needs to be reduced to limit nonspecific binding. It may also be possible to photobleach or otherwise monitor the fluorescence intensity of the specific versus nonspecific interactions to determine whether more fluorescent proteins are bound to certain sequences.

We thank the reviewer for these comments and questions. Following the reviewer's suggestion, we conducted new control experiments. For EWS-FLI1, we found that the fluorescent signals were photobleached less in the region of 25× GGAA repeats than the region without 25× GGAA repeats (Response Fig. 1.2a-b), indicating that more fluorescent proteins were bound to the region of 25× GGAA repeats. As for FUS-Gal4, when the concentration of mCherry-FUS-Gal4 was decreased to 10 nM (Response Fig. 1.2c), as expected the nonspecific binding events were indeed limited, and most of mCherry-FUS-Gal4 molecules bound at the specific binding sequence.

In addition, we would like to clarify further on the experimental condition of DNA Curtains that have been used in this study. In Fig. 3, we injected dark FUS-Gal4 (without a fluorescent tag) into the chamber. Before injection, we used YOYO-1 to stain DNA substrates. Therefore, "the widespread fluorescence in Figure 3Bi and Figure 3Fi", mentioned by the reviewer, were the signals of YOYO-1/DNA, not the signals from dark FUS-Gal4. After the injection of FUS-Gal4, green-colored YOYO1

puncta appeared on DNA (Fig. 3b(i), (ii), and h(i)), suggesting that FUS-Gal4 associated with the DNA through a mechanism that gave rise to regions of local high DNA-binding site concentrations. Furthermore, we show that the position distribution of these YOYO1 puncta closely coincided with the insertion position of 7× Gal4DBD binding sites (Fig. 3b(ii)), suggesting the specific interactions between FUS-Gal4 condensates and the 7× Gal4DBD binding site.

Response Fig. 1.2 | The control experiments to confirm FET fusion proteins binding at their specific binding sites. (a) Wide-field TIRFM images of one representative DNA substrate after 100 nM EWS-FLI1 injection (Fig. 2c(ii)). A 561-nm laser was used and the exposure time was 100-ms. Each snapshots represent the total exposure time from 0 to 15 seconds. (b) The normalized intensities of the 25×

GGAA repeats region (red) and non-25× GGAA repeats region (black) in (a) was analyzed versus the total exposure time. The error bars represent s.d. (c) Wide-field TIRFM images of DNA Curtains for 10 nM mCherry-FUS-Gal4 binding at Lambda DNA containing 7× Gal4DBD binding sites. (i) Flow on; (ii) Flow off.

2. It would be ideal to show that the phase separation occurs without the GFP or mCherry tags (Figure 1), which can lead to increased solubility or dimerization. The untagged droplets should be morphologically similar to those formed by the tagged fusion proteins. It would also be helpful to show that the liquid-like properties are not affected by performing FRAP with fluorescently-labeled DNA (e.g. A408 or Cy5 label or some other label that does not undergo FRET with GFP or mCherry) droplets for the untagged and GFP-/mCherry-tagged fusion proteins. These experiments are necessary to show that GFP and mCherry tags do not affect the fusion proteins' LLPS properties.

We thank the reviewer for these comments and questions. We purified dark FUS-Gal4 and SNAP-EWS-FLI1 without the fluorescent tags, and monitored the activity of these dark fusion proteins using a differential interference contrast (DIC) microscopy (Response Fig. 1.3). We found that the untagged droplets were morphologically similar to those formed by the tagged fusion proteins (Fig. 1d and j).

In addition, we conducted the FRAP experiments with Quasar670-labeled DNA in Response Fig. 1.3 and also Fig. 1e and 1j. However, the fluorescent Quasar670 signals of DNA were not recovered after photobleaching. This is consistent to what we and others¹²⁻¹⁴ have found that longer nucleic acids in the biomolecular condensates are rarely recovered after photobleaching, which is an interesting phenomenon warrant future studies.

Response Fig. 1.3 | Dark FET fusion proteins can form biomolecular condensates with DNA containing DNA binding sites. (a) 16 μM FUS-Gal4 (no PEG8000) mixed with 0.36 μM 326-bp dsDNA. (i) Differential interference contrast (DIC) channel; (ii) Quasar670 channel; (iii) Merge. DNA substrates contained 11x Gal4DBD binding sites and labeled with Quasar 670. (b) 15 μM SNAP-EWS-FLI1 (no PEG8000) mixed with 0.45 μM 306-bp dsDNA. (i) DIC channel; (ii) Quasar670 channel; (iii) merge. DNA substrates contained 25x GGAA repeats and labeled with Quasar 670.

3. There is limited evidence that the punctae in Figures 2-4 and 6 are actually liquid-like. The authors note that they sometimes observe fusion events, but these should be quantified and included in the main Figures alongside the data noting the existence of the punctae.

We thank the reviewer for this suggestion. We included the fusion events in Supplementary video S3 in Fig. 3b(iii), and quantified them in Fig. 3b(iv) with the revised main text at Page 8.

4. The authors also note that a higher number of GGAA repeats causes increased transcription in Figure 5. However, the in vivo luciferase experiments do not definitively show that the fusion protein is responsible for the increase in transcription of these longer GGAA repeats. Another factor within their cell line could easily be reading these repeats and promoting transcription above the same threshold repeat length (which appears to be 5 or 6 in Figures 5A-C). The authors need to perform a control without a fusion protein co-transfected to show that there is either a shift in the number of repeats needed for transcription or that the intensity of transcription is higher because of the fusion protein co-transfection. Moreover, the ChIP-Seq data from Figures 5D-G only weakly corroborates the notion that transcription increases with GGAA repeat length (especially Figures 5E/G; Figure 5F is not significant). The way to further stress the importance of the GGAA repeat length is to repeat the experiments from Figures 3-4 with the fusion proteins but vary the length of the GGAA repeat. If a difference in the apparent transcription scales with the GGAA repeat length, this would further corroborate the in vivo evidence shown in this figure and lend weight to the assertion that the fusion proteins – and not something else – are responsible for this increased transcription.

We thank the reviewer for these comments and questions. Following the reviewer's suggestion we performed the control experiment without co-transfection of the fusion proteins (Fig. 5d). There was little luciferase activity observed without the fusion protein, further supporting that the fusion protein is responsible for the increased

transcription of these longer GGAA repeats. We updated the relevant main text at Page 14.

We agree with the reviewer's comment about the CHIP-Seq data analyses. The third reviewer has the similar comment (Page 45 in this document), thus we move these analyses to the supplementary part (Supplementary Fig. 8).

To further address the importance of the DBD binding site repeat length, we conducted DNA Curtains experiments (Response Fig. 1.4). First, we discussed FUS-Gal4. we used wild type Lambda DNA (without our designed Gal4DBD binding sites) as DNA substrates in the *in vitro* transcription assay (Fig. 4d) showing a very low transcription efficiency (0.05, N = 278, Lane 4 in Fig. 4f). Second, we discussed EWS-FLI1. We performed new DNA Curtains experiments to study EWS-FLI1: (i) EWS-FLI1 can also recruit Pol II CTD (Supplementary Fig. 6a and d, and Supplementary video S5); (ii) The recruitment of Pol II CTD by EWS-FLI1 condensates can activate gene transcription around the target loci (Fig. 4g or Response Fig. 1.4a here). When we used the wild type Lambda DNA (the consecutive motif is only 2× GGAA, Response Fig. 1.4b) as DNA substrates in the *in vitro* transcription assay, no nascent RNA transcripts were monitored (Response Fig. 1.4c). Taken together, these data suggested that the DNA binding motif might be another important factor controlling *in vitro* transcription efficiency.

Response Fig. 1.4 | The recruitment of RNA polymerase by EWS-FLI1 condensates promotes gene transcription activity *in vitro*. (a) Same as Fig. 4g. Schematic of *in vitro* single-molecule biomolecular condensate-induced transcription assay for EWS-FLI1 (i). Wide-field TIRFM images of nascent RNA transcripts (iii) colocalized with EWS-FLI1 condensates (ii) after incubation. White arrows in (iii) confirmed the labeled punctum was on DNA. DNA substrates were Lambda DNA containing 25x GGAA repeats. (b) Line plot showing the hypothetical distribution of

the number of puncta that would be visible if EWS-FLI1 recruitment was driven by a threshold number of consecutive motifs. (c) The control experiment of (a): DNA substrates were wild type Lambda DNA (no 25× GGAA repeats). The working buffer was 40 mM Tris-HCl (pH 7.5), 25 mM KCl, and 2 mM MgCl₂, 1 mM DTT, and 0.2 mg/ml BSA.

5. In Figure 3F, it is unclear whether this interaction between the Pol II FUS is actually specific or not. When the flow is turned off (Figure 3Fiii), the Pol II seems to preferentially localize to the top of the DNA strand away from the Gal4 DBD binding sites. As the authors show that the Pol II CTD can diffuse into and out of the condensates in Figure 3D-E, it seems like the Pol II CTD is diffusing out of the condensates in Figure 7Fii and preferentially interacting with another part of the DNA strand in Figure 7Fiii. It may be possible that the Pol II CTD is only localizing with the condensate when the flow is on because of the higher local viscosity of the condensates trapping the Pol II CTD rather than through a specific interaction. The low in vitro transcription observed in Figure S4J observed with physiological salts further supports the notion that this interaction is exceedingly weak. The authors should test for a specific interaction between FUS and the Pol II CTD and whether the binding between FUS, the Pol II CTD, and the DNA is cooperative.

We thank the reviewer for this valuable question. We feel that we should have explained more clearly about DNA Curtains method used in this study as follows, which might be helpful to answer the reviewer's question here. In a typical DNA Curtain set up, one end of a DNA substrate is tethered to an individual lipid in the supported lipid bilayer within a microfluidic chamber by a biotin-streptavidin interaction^{15,16}. The working buffer flow can extend DNA substrates. When the flow is off, DNA molecules

will shrink back to the barriers, and all binding events present on DNA will also move with DNA. On the other hand, all molecules that are stuck on the flow cell surface cannot move with DNA at the same time when the flow is off. This is an important benefit of DNA Curtains method, which can distinguish these binding events on DNA from those molecules stuck on the single-molecule surface.

In the present study, we found enriched mCherry signals representing Pol II CTD_{N26}-mCherry in Fig. 3h(iii) when the flow was on. These signals that were labeled with white arrows disappeared when the flow was off (Fig. 3h(iv)), indicating these signals indeed bound to DNA and did not stick on the single-molecule surface. The signals from the green channel (Fig. 3h(i)-(ii)) also confirmed that DNA substrates extended and shrank on the same time when the flow was on and off, respectively. Therefore, our data do not show that “the Pol II CTD is diffusing out of the condensates in Fig. 3F(iii) and preferentially interacting with another part of the DNA strand in Fig. 3F(iii).”. We also clearly labeled the barrier positions on all wide-field TIRFM images in Fig. 3h, and Supplementary video S4 was captured to show this process clearly.

Minor Concerns

6. The authors' title states that this is “aberrant” gene transcription – but we have no way to know whether this is aberrant or not given that there are no controls with either half of the fusion protein. Given that the vast majority of experiments use an artificial FUS-Gal4 system and a Gal4 DNA binding element, there is no way to conclude that this is an aberrant interaction because it is artificial.

We thank the reviewer for this comment, and revised the title as “Loci-specific phase separation of FET fusion oncoproteins promotes gene transcription”.

7. It is unclear why Figure S2A and Figure S2C are included in the same figure as Figure S2B and Figure S2D. These should be separated.

We thank the reviewer for this question. We are sorry for the confusion made regarding to these figures. Figure S2A and Figure S2B in the old manuscript are two steps in one experiment, so as Figure S2C and Figure S2D (in the old manuscript), following the same schematic in Fig. 3a.

In the new manuscript, the old Figure S2 is named as Supplementary Fig. 4. To make it clearer, we adjusted the figures accordingly. Our main experiment confirmed that treatments with 150 nM unlabeled FUS-Gal4 yielded YOYO1 puncta on DNA (Fig. 3b(i)), and these YOYO1 puncta are able to recruit Pol II CTD to the fusion binding motif (Fig. 3h). We repeated this experiment with two different experimental conditions: (i) Supplementary Fig. 4a demonstrated 10 nM FUS-Gal4 cannot generate YOYO1 puncta on DNA (Supplementary Fig. 4a(i)), and thus no Pol II CTD can be recruited at the fusion binding motif (Supplementary Fig. 4a(ii)-(iv)); (ii) Supplementary Fig. 4b demonstrated 150 nM Gal4DBD is not able to initiate YOYO1 puncta on DNA (Supplementary Fig. 4a(i)), and thus no Pol II CTD was recruited at the fusion binding motif (Supplementary Fig. 4b(ii)-(iv)).

8. The data from Video S2 is not quantified or shown in the main figures, and it should be included in a main or supplemental figure. Otherwise, it is difficult to evaluate the authors' claim that there is a significant increase in the green channel intensity upon addition of YOYO-1.

We thank the reviewer for this comment. As suggested, we generated the new Supplementary Fig. 3 to show Supplementary video S2. We quantified the green

channel intensity before (Supplementary Fig. 3b(i)) and after (Supplementary Fig. 3d(i)) YOYO-1 injection, and found that there is a significant increase in the green channel intensity upon addition of YOYO-1 (Supplementary Fig. 3e).

9. For the sake of consistency, the mCherry labeled images in Figures 3c-e should be colored magenta like other mCherry constructs.

We thank the reviewer for this comment. As suggested, we synchronized the colors in Fig. 3c-e. Similarly, we also changed the color of Quasar 670 in Fig. 1d and j from magenta to red. All main figures related to the mCherry tag are now showing in magenta.

10. The figures could be more clearly labeled to enable easier reading of the data. For instance, in many panels (Figures 1E, 1J, 3B, 3C, 3F, 4B, 6B, 6E, and 6G it is not clear what is being fluorescently labeled and what is being visualized simply by looking at the figure.

We thank the reviewer for this comment. As suggested, we updated all labels in the main figures and related Supplementary Figures.

11. In general, more of the controls should be included in the main figures. Since the authors are using the DNA curtain assay and there is still a lot of room in many of these figures (especially Figure 4), it would make the paper more readable to include these controls and experiments in the main figures when appropriate. The figures are especially lacking in images of the control conditions in the main figures; for instance,

Figure 4 should include sample images of all three lanes shown in the quantification (Figure 4C).

We thank the reviewer for this comment. As suggested, we added all control conditions in Fig. 4. We also followed the reviewer's suggestion to check all other main figures, like Fig. 1, 2, 3, 5, and 6, and add all control experiments needed.

Reviewer #2 (Remarks to the Author):

In this article, the authors present a direct test of the causal links between condensates formed by fusion oncoproteins and increased transcription. This is an idea that has been proposed and discussed in the literature. Here, the authors deploy an in vitro, single molecule transcription assay, and show that the transcriptional output is directly tied to (i) condensate formation along the DNA; (ii) recruitment of RNA Pol II; (iii) and efficient and accurate generation of RNA transcripts that is governed, in part, by the valence of specific DNA motifs known as microsatellites.

The collection of data presented in this work are impressive, persuasive, and likely to garner significant interest within the readership of Nature Communications. Being able to demonstrate causal links, especially in vitro, will help place the field of biomolecular condensates on firmer footing. This MS goes a long way toward making this happen. It is well written, is quite comprehensive, and nothing is really left to chance. There are few points that would benefit from being addressed in a revised version.

We thank the reviewer for these important comments and questions, and conducted all new control experiments (see below). We have revised the relevant text, prepared new Supplementary Figures, and cited all relevant references as suggested in the new manuscript. All changes in the manuscript text file were shown with red color highlighting.

These are as follows:

1) The in vitro phase behavior of EWS-Fli1 and FUS-Gal4 raise an important question: These systems undergo phase separation at concentrations that are in line with the regimes established by Wang et al., 2018. However, instead of the RNA binding

domains, we now have DNA binding domains. It would help if the authors were to discuss commonalities between the DBDs used here and the RBDs used in the molecular grammar work of Wang et al.

We thank the reviewer for this great suggestion. Many previous references found that the low complexity domains (LCDs) of FET needed much higher concentrations to form droplets ^{8,9}. For example, Fawzi and coworkers found that FUSLCD can form droplets at 200 μ M under certain experimental conditions, such as low temperature and high salt *in vitro* ⁸. In contrast, when RNA-binding domains (RBDs) were added to form a full length version of FET (FUS, EWSR1, or TAF15), the droplets can be formed at a much lower concentration ⁹. Based on these findings, Alberti, Pappu and coworkers identified a molecular grammar of phase separation ⁹, which is the study (Wang *et al.*) suggested by the reviewer. They found that Tyrosine residues from the LCD can form multivalent interactions with Arginine residues from the RBD.

In our study, the FET fusion proteins contain DNA binding domains (DBDs) instead of RBDs. We found that when DBDs were connected to LCDs, the FET fusion proteins, such as FUS-Gal4 and EWS-FLI1 can also form droplets at a very low concentration of 2 μ M (Fig. 1a and g). It strongly suggested that LCD-DBD interactions may contributed to the formation of FET fusion protein condensates. Studies by Lessnick and coworkers ¹ also confirmed the LCD-DBD interactions. Therefore, it seems that there are some commonalities between the DBDs used in our study and the RBDs used in the molecular grammar work by Wang et al. ⁹. Here, we designed and conducted new control experiments to further test the commonalities, and the new results were shown in Response Fig. 2.1 below.

We first followed the previous studies ^{9,17} to mutate all Tyrosine residues in 27 [G/S]Y[G/S] tripeptide repeats of FUS LCD to Serine residues, and this mutant was

referred as GFP-FUS(Y-S)-Gal4. In comparison to GFP-FUS-Gal4 (Response Fig. 2.1a, or Fig.1a), GFP-FUS(Y-S)-Gal4 cannot form droplet (Response Fig. 2.1b), suggesting that the interactions between Tyrosine and Arginine can drive phase separation of FET fusion proteins. This control experiment suggested that the molecular grammar identified by Wang et al.⁹ also applies to FET fusion proteins.

Secondly, we tested GFP-EWS-FLI1(R2L2), in which Arginine residues were mutated within the FET DBD (R383L and R386L) with decreased the DNA binding ability compared to the wild type FLI1DBD¹⁸. In comparison to GFP-EWS-FLI1 (Response Fig. 2.1c, or Fig.1g), GFP-EWS-FLI1(R2L2) has much weak ability to form biomolecular condensates (Response Fig. 2.1d), suggesting the Arginine residues from the FET DBD might also affect the biomolecular condensates formation.

Taken together, the multivalent interactions between Tyrosine residues from FET LCD and Arginine residues from FET fusion protein DBDs can drive phase separation of FET fusion proteins. This is an interesting point warrant for future studies. We revised the relevant main text at Page 19-20.

Response Fig. 2.1 | Molecular grammar of phase separation for FET fusion proteins. (a-d) *In vitro* droplet experiments for GFP-FUS-Gal4 (a), GFP-FUS(Y-S)-Gal4 (b), GFP-EWS-FLI1 (c), and GFP-EWS-FLI1(R2L2) (d). (i) 2 μ M; (ii) 5 μ M; (iii) 10 μ M; (iv) 20 μ M; (v) 30 μ M. (a) is the same data of Fig. 1a, and (c) is the same data of Fig. 1g.

2) The methods and the figure captions do not provide information regarding the salt concentrations at which *in vitro* condensates are formed by the fusion proteins. One would have to presume that these condensates dissolve at monovalent salt concentrations of ca. 150 - 250 mM, and are stabilized at lower salt concentrations such as 75 mM NaCl. Is this correct? If so, might it be the case that the loss of

transcriptional activity at 150 mM NaCl is due to the dissolution of condensates? An explanation of this issue would be very helpful.

We thank the reviewer for these comments and questions. We are sorry that we did not describe the salt condition clearly. We revised the relevant text (Methods and figure captions) to better describe the experimental condition. All *in vitro* droplet assays, EMSAs, and most of DNA Curtains experiments were conducted at monovalent salt (KCl) concentrations of 150 mM. Only two DNA Curtains experiments were conducted by using the low KCl salt concentrations: (i) FUS-Gal4 condensates only slightly compacted the DNA substrates, even in the case of low salt concentration (<25 % for 10 mM salt) (Supplementary Fig. 5c); (ii) We also found that the salt concentration affected the *in vitro* transcription efficiency, and 150 mM KCl caused a much lower transcription efficiency than 25 mM KCl (Fig. 4b and Lane 1 in Fig. 4f).

Whether FUS-Gal4 condensates can easily dissolve at 150 mM KCl or not? To answer this question, we conducted new kinetic measurements of the condensate-DNA interaction at 150 mM KCl (Response Fig. 2.2a-b). The data revealed that the droplet association (Response Fig. 2.2c) exhibits characteristic power-law behavior, reflecting the existence of a diverse ensemble of transient complexes¹⁹. The on-rate, k_{on} , was measured to be $(4.8 \pm 0.2) \times 10^{-5}$ (second⁻¹·nM⁻¹) (Response Fig. 2.2c), and the off rate, k_{off} , to be $(37.6 \pm 1.5) \times 10^{-6}$ second⁻¹ (Response Fig. 2.2d), with most puncta still bound to DNA Curtains after the two-hour measurement period. The dissociation constant, $K_D (= k_{off} / k_{on})$, is 0.8 ± 0.5 nM for the condensate-DNA interaction. These data proved that FUS-Gal4 condensates can bind stably to DNA at least two hours. However, the total time of our *in vitro* transcription assay only took around 30 minutes (Fig. 4a). Thus, the loss of transcriptional activity at 150 mM KCl (Lane 1 and 2 in Fig. 4f) is not due to the dissolution of condensates. As the FUS-Gal4

condensates can compact DNA more in the case of lower salt concentration (Supplementary Fig. 5c), we assumed that the FUS-Gal4 condensates can contain more FUS-Gal4 molecules at 25 mM KCl than at 150 mM, and then the condensates at lower salt condition can recruit more RNA polymerases to active gene transcription. We conducted the control experiment to test this assumption for EWS-FLI1, and EWSFLI1 molecules indeed can recruit more RNA polymerases at 25 mM KCl than at 150 mM KCl (Supplementary Fig. 6a, b and d).

Response Fig. 2.2 | kinetic measurements of the FUS-Gal4 condensate-DNA interaction at 150 mM KCl. (a-b) Representative kymograms showing association (a) and dissociation (b) of green puncta on DNA. **(c-d)** Association (c) and dissociation (d) kinetics of LLPS. Error bars for survival probability were obtained through bootstrap analysis.

3) The schematic and the overall mechanism that emerge could use some clarification. Do the authors propose that condensates form autonomously and dock onto DNA by displaying a multivalence of DBDs? Or do the condensates form around the DNA

leveraging a combination of homotypic interactions (presumably involving the LCD and DBD) and heterotypic interactions (mainly involving the DBD and DNA but also the LCD and DNA) whereby the DNA weaves through condensates? The schematic in Fig. 7 appears to be consistent with both mechanisms and whether or not the data can adjudicate between these models is unclear. What would help is the interrogation of the phase behavior of either EWS-Fli1 or FUS-Gal4 in the presence of oligos comprising the GGAA motifs. This is a relatively straightforward experiment whereby one can titrate the concentration of GGAA motifs along one axis and the concentration of oncoprotein along the other axis and assess the contribution of motif interactions to condensate formation. If the phase boundary shows a strong synergy between the two concentrations, then there is an interplay of homotypic and heterotypic interactions and the model is a weave through rather than a dock onto DNA model.

We thank the reviewer for these great suggestions. To answer the reviewer's question, we followed the reviewer's suggestion to design and conduct new experiments (Supplementary Fig. 10). GFP-FUS-Gal4 formed small droplets at 8 μ M (Supplementary Fig. 10a(i)). Mixing GFP-FUS-Gal4 with DNA including varying numbers of Gal4DBD binding sites resulted in larger droplets (Supplementary Fig. 10a and b). The Gal4DBD alone did not form droplets without (Supplementary Fig. 10b) or with DNA (Supplementary Fig. 10c). Phase separation diagrams of protein and DNA (Supplementary Fig. 5e-g) revealed: (i) the presence of DNA promoted the FUS-Gal4 condensates by lowering the critical concentration for FUS-Gal4 phase separation; and (ii) the presence of more Gal4DBD binding sites on the DNA resulted in even lower critical concentration of FUS-Gal4. The phase boundary indeed showed a strong synergy between the concentration DNA binding motifs and the concentration of FET fusion protein. These results strongly suggested that the FET fusion protein

condensates can form around the DNA leveraging a combination of homotypic interactions (presumably involving the LCD and DBD of FET fusion protein) and heterotypic interactions (mainly involving the DBD and DNA) whereby the DNA weaves through condensates. As we already confirmed GFP-FUSLCD (Supplementary Fig. 2i) cannot bind to the 1× Gal4DBD sequence and also the control sequence, the LCD and DNA cannot form the heterotypic interactions mentioned above. We revised the relevant main text at Page 18-19.

Once these issues have been clarified, this MS should be acceptable for publication, at least from my perspective.

Reviewer #3 (Remarks to the Author):

In this manuscript, “Loci-specific phase separation of FET fusion oncoproteins promotes aberrant gene transcription,” Zuo et al. purify recombinant FET fusion proteins and evaluate the ability of these fusion proteins to phase separate and recruit the RNA polymerase II C-terminal domain in vitro using a DNA curtains assay. They also develop a novel in vitro transcription assay based upon the DNA curtains technique to evaluate the mechanistic link between phase separation and local transcriptional activation. The main claims of the paper are that they have developed a novel in vitro single molecule assay for DNA-templated phase separation, that FET fusion proteins undergo phase separation at target loci, that phase separated condensates recruit RNA polymerase II and enhance transcription, and that they determined a threshold of fusion-binding DNA elements.

Zuo et al. have taken an innovative approach to address important questions regarding the contribution of phase separation to oncogenic FET-ETS fusion transcription factor function. It is widely speculated that phase separation at specific loci is a critical facet of FET-ETS function in pediatric tumors, but the evidence published to date is circumstantial. Direct evidence would represent a step forward for the field. This paper has several strengths conferred by using the DNA curtains assay, both as a stand-alone assay and as the basis for an in vitro transcription assay. The most striking finding of the paper is that recruited T7 polymerase initiates transcription even when recruited by a FUS condensate in the absence of a T7 promoter. The authors also spent a significant amount of time deeply considering the existing literature for EWS/FLI binding at GGAA microsatellites and worked to thoughtfully incorporate some of the more subtle pieces of the literature into the interpretation of their results. Unfortunately, this paper also had several key weaknesses that undercut

the ability to meaningfully apply the data gathered here to EWS/FLI function. These include a lack of controls required to interpret the EWS/FLI experiments, EWS/FLI experiments that appear to not support the model put forward, and collection of data primarily from an artificial FUS-Gal4 construct. While the in vitro transcription finding is quite interesting and compelling, this manuscript cannot be published in its current form without addressing these weaknesses, outlined in the comments to authors below.

We thank the reviewer for these important comments and questions, and conducted all new control experiments (see below). We have revised the relevant text, prepared new Supplementary Figures, and cited all relevant references as suggested in the new manuscript. All changes in the manuscript text file were shown with red color highlighting.

Major concerns:

EWS/FLI is a notoriously difficult protein to purify and work with. A Coomassie gel showing the purification of recombinant EWS/FLI and FUS-Gal4 used for experiments should be shown in the Supplement or Extended Data. This is required to show that the effects seen across all experiments are really coming from the purified protein and not an artifact of the inclusion of some impurity in the protein prep.

We thank the reviewer for these suggestions. We followed the reviewer's suggestion showed the SDS-PAGE gels for FUS-Gal4 and EWS-FLI1 in Supplementary Fig. 1.

In a similar vein, the EMSA assays in Supplementary Figure 1 all suggest the authors have, indeed, purified functional protein, with the exception of panels f and g. These

are lacking a shifted band at a higher molecular weight, and instead only show the loss of the lower band.

We thank the reviewer for these comments and questions. We repeated all EMSA experiments in new Supplementary Fig. 2. In comparison to the old EMSA, we firstly added DNA substrates containing no FET fusion protein binding sites as control. Secondly, in order to achieve better imaging results, DNA substrates were labeled with Quasar670, and imaged by Amersham Typhoon RGB (with a 635 nm laser and Cy5 670BP30 filter). This method can effectively reduce the noise signals from GFP or mCherry tag in EMSA. Thirdly, we used a lower concentration of PAGE gel (8%) instead of 12%, so that we could observe more clearly shifted bands, especially for new Supplementary Fig. 2g and h. Here new Supplementary Fig. 2g is the old Supplementary Figure 1f, and new Supplementary Fig. 2h is the old Supplementary Figure 1g. Taken together, our new EMSA experiments demonstrated that EWS-FLI1 and FUS-Gal4 specifically bind to their known target sequences used in this study. We revised the relevant main text at Page 5 of the new manuscript.

Figure 1 is currently uninterpretable without additional controls, including GFP alone. An appropriate negative control showing no phase separation with the tag is important, as well as the GFP-DBD (i.e. GFP-Gal4 and GFP-FLI) construct to show that the intrinsically disordered N-terminus of FUS and EWS is required for phase separation as predicted by the overarching model. Moreover, it's important to know whether or not DNA phase separates into droplets because of the binding affinity of the DBD in the droplet, or simply because of physical factors. Appropriate controls for this include

a DNA-binding mutant (such as the R2L2 mutant of FLI) or DNA lacking the binding motif (i.e. GGAT microsatellites or scrambled DNA).

We thank the reviewer for these comments and questions. As suggested, we conducted all new control experiments as mentioned by the reviewer (Response Fig. 3.1). First, even 100 μ M GFP (Response Fig. 3.1a(i)) or 100 μ M mCherry (Response Fig. 3.1a(ii)) alone cannot form biomolecular condensates. The GFP molecule fused in all constructs in this work was the superfolder GFP (sfGFP), which contained an single mutation (A206K) and can avoid the weak dimerization of GFP²⁰. Response Fig. 3.1a(i) is the same as Fig. 1b(ii).

Second, in comparison to 30 μ M GFP-FUS-Gal4 (Response Fig. 3.1b(ii)) and GFP-EWS-FLI1 (Response Fig. 3.1c(ii)), 30 μ M GFP-Gal4DBD (Response Fig. 3.1b(i)) or GFP-FLI1DBD (Response Fig. 3.1c(i)) cannot form biomolecular condensates. It suggests that the intrinsically disordered N-terminus of FET is required for phase separation.

Third, we followed the reviewer's suggestion to purify GFP-EWS-FLI1(R2L2) *in vitro*. It contains two mutants, R383L and R386L, which decreases the DNA binding ability of FLI1DBD¹⁸. From EMSA assays, we confirmed again that GFP-EWS-FLI1(R2L2) (Supplementary Fig. 16c) cannot bind to the microsatellite DNA. For the *in vitro* droplet assay, it seems like GFP-EWS-FLI1(R2L2) can only form fiber, which was completely different from the liquid-like biomolecular condensate of GFP-EWS-FLI1 (Response Fig. 3.1c(ii)). Thus, the *in vitro* droplet assay was not a good method to answer the reviewer's question: "whether or not DNA phase separates into droplets because of the binding affinity of the DBD in the droplet, or simply because of physical factors."

To answer this question, we conducted new DNA Curtains experiments in Fig. 2. In comparison to 500 nM mCherry-EWS-FLI1 (Fig. 2c), 500 nM mCherry-EWS-

FLI1(R2L2) cannot form biomolecular condensates on the locus at the location of the 25× GGAA repeats (Fig. 2I). Thus, we confirmed that DBD can recruit DNA into the biomolecular condensates.

We revised the relevant main text at Page 5 of the new manuscript.

Response Fig. 3.1 | The control experiments of *in vitro* droplet assays for Fig. 1.

(a) 100 μ M GFP (i) (Fig. 1b(ii)); 100 μ M mCherry (ii). (b) 30 μ M GFP-Gal4DBD (i) and 30 μ M GFP-FUS-Gal4 (ii) (Fig. 1a(v)). (c) 30 μ M GFP-FLI1DBD (i) and 30 μ M GFP-EWS-FLI1 (ii) (Fig. 1g(v)). (d) ESMA for GFP-EWS-FLI1(R2L2). (e) 55 μ M GFP-EWS-FLI1(R2L2).

Figure 2 is currently uninterpretable without additional controls. These controls are very similar to those discussed above for figure 1 and include an mCherry alone, an

mCherry-FLI DBD construct (data is included, but with no DNA curtains figure), an mCherry-EWS construct, and a DNA-binding mutant of EWS/FLI (i.e. R2L2). These constructs all need both a DNA curtains figure and the puncta intensity included in the plot. Alterations to the DNA sequence (i.e. GGAT repeats) would also be helpful.

We thank the reviewer for these suggestions. As suggested, we conducted all control experiments and updated Fig. 2. First, we included DNA Curtains data for mCherry-FLI1DBD from 500 nM, 100 nM to 20 nM into Fig. 2 (Fig. 2f-h). Second, we included the new results of DNA Curtains for mCherry alone (Fig. 2i), mCherry-EWSLCD (Fig. 2j), and mCherry-EWS-FLI1(R2L2) (Fig. 2k). GFP-EWS-FLI1(R2L2) contains two mutants, R383L and R386L, which decreases the DNA binding ability of FLI1DBD¹⁸. These control experiments clearly showed that no magenta puncta appeared on the locus at the location of the 25× GGAA repeats. We revised the relevant main text at Page 6-7.

Figure 2 does not show a single puncta at the microsatellite and instead shows EWS/FLI coating the DNA strand. The puncta at the microsatellite do not appear to be more intense than the puncta elsewhere, somewhat negating the discussion of the behavior focusing on microsatellite (page 6, second paragraph). Does the YOYO-1 after EWS/FLI infusion match the EWS/FLI puncta throughout the DNA strand?

We thank the reviewer for these comments and questions. As suggested, we first conducted a control experiment to compare the puncta intensity at the microsatellite with the puncta intensity elsewhere. We found the fluorescent signals were photobleached less in the region of 25× GGAA repeats than the region without 25×

GGAA repeats (Response Fig. 3.2a-b), indicating that more fluorescent proteins were bound to the region of 25× GGAA repeats.

Second, we conducted DNA Curtains experiments (Response Fig. 3.2c). After 250 nM mCherry-EWS-FLI1 was injected into the flow cell, when the 488-nm laser was turned on to image YOYO-1 for DNA, green-colored YOYO1 puncta appeared on DNA (Response Fig. 3.2c(i)-(ii)). When we switched to only turn on the 561-nm laser for imaging mCherry-EWS-FLI1, high-intensity magenta puncta of EWS-FLI1 appeared on DNA (Response Fig. 3.2c(iii)-(iv)). We found that the YOYO-1 after EWS-FLI1 infusion match the EWS-FLI1 puncta throughout the DNA strand.

Response Fig. 3.2 | The control experiments to confirm FET fusion proteins binding at their specific binding sites. (a) Wide-field TIRFM images of one representative DNA substrate after 100 nM mCherry-EWS-FLI1 injection (Fig. 2c(ii)). A 561-nm laser was used and the exposure time was 100-ms. Each snapshots represent the total exposure time from 0 to 15 seconds. (b) The normalized intensities of the 25x GGAA repeats region (red) and non-25x GGAA repeats region (black) in (a) was analyzed versus the total exposure time. The error bars represent s.d. (c)

Wide-field TIRFM images of DNA Curtains after 250 nM mCherry-EWS-FLI1 injection. (i)-(ii) only the 488-nm laser on to image YOYO1 for DNA and flow on (i) / off (ii); (iii)-(iv) only the 561-nm laser on to image mCherry for mCherry-EWS-FLI1 and flow on (iii) / off (iv).

For Figures 3 and 4 (and supporting Extended Data), the authors use a FUS-Gal4 construct similar to that used by the McKnight lab. As for figure 1, negative controls with just the GFP and GFP-Gal4 need to be included in the phase separation assay with Pol II CTD.

We thank the reviewer for this suggestion. As suggested, we conducted these control experiments. The data showed that just the GFP (Fig. 3f) or GFP-Gal4DBD (Fig. 3g) cannot form biomolecular condensates with Pol II CTD.

Figures 3 and 4 show some interesting phase separation activity, Pol II CTD recruitment, and in vitro transcription. However, these data are not generally applicable to EWS/FLI as the authors have written. First, there are distinct differences in the number and density of critical (G/S)Y(G/S) repeats in FUS and EWS (and TAF15) that may have important biological relevance. Second, Gal4 and ETS transcription factors bind DNA in completely different manners. Gal4 homodimerizes such that 2 Gal4 proteins bind a single Gal4 motif, while ETS family member bind monomerically. Gangwal et al. (2010 Genes Cancer) and Johnson et al. (2017 PNAS) show that a single EWS/FLI binds monomerically to a 2X GGAA repeat. Moreover, Johnson et al. has recently suggested that the EWS domain of EWS/FLI alters the DNA-binding

function of FLI to confer the ability to bind GGAA microsatellites, such that binding multiple EWS/FLI molecules at a microsatellite is mechanistically different than simply stringing together multiple high affinity sites, as is done here for Gal4. Unless the appropriate controls are used to evaluate the difference between multiple high affinity ETS sites and GGAA microsatellites with EWS/FLI the authors should revise portions of their paper where they apply finding from FUS-Gal4 to EWS/FLI. A more appropriate comparison might be EWS/ATF1 or EWS/ATF7 where the DBD of the fusion is also from a transcription factor which dimerizes to bind DNA.

We thank the reviewer for these great comments and questions. As suggested, we conducted new experiments to answer the reviewer's questions. First, we performed new DNA Curtains experiments to directly study EWS-FLI1: (i) EWS-FLI1 can also recruit Pol II CTD (Supplementary Fig. 6a and d, and Supplementary video S5); (ii) The recruitment of Pol II CTD by EWS-FLI1 condensates can activate gene transcription around the target loci (Fig. 4g).

Second, we followed the reviewer's suggestion to study EWS-ATF1 (Response Fig. 3.3). We purified GFP-EWS-ATF1 *in vitro* (Response Fig. 3.3a), and EMSA was conducted to confirm that there is a specific DNA binding interaction between GFP-EWS-ATF1 and DNA (Response Fig. 3.3b). We conducted the *in vitro* droplet assays to confirm that GFP-EWS-ATF1 can form biomolecular condensates without or with DNA (Response Fig. 3.3c-d). From the luciferase assays, we also found that ATF1DBD binding motifs beyond a threshold number can drive the formation of EWS-ATF1 condensates (Response Fig. 3.3e). The scale of the induced luminescence of EWS-ATF1 is similar as FUS-Gal4 (Fig. 5a). When we repeated DNA Curtains experiments for EWS-ATF1, we found that GFP-EWS-ATF1 can form biomolecular condensates at the 11× ATF1DBD binding loci (Response Fig. 3.3f). The condensates

can also recruit Pol II CTD (Response Fig. 3.3g), and the recruitment of Pol II CTD by EWS-ATF1 condensates can also activate gene transcription around the target loci (Response Fig. 3.3h).

Taken together, we confirmed that all these FET fusion proteins, including EWS-FLI1, and EWS-ATF1, and also our model system FUS-Gal4, show the same activities, like phase separation activity, Pol II CTD recruitment, and *in vitro* transcription. We revised the relevant main text at Page 3, 10, 12, and 13, and also cited the relevant references suggested by the reviewer.

Response Fig. 3.3 | EWS-ATF1 has the same results as EWS-FLI1 and FUS-Gal4.

(a) SDS-PAGE for GFP-EWS-ATF1. (b) EMSA for GFP-EWS-ATF1. (i) Schematic; (ii)

EMSA (8% native PAGE gel). We chose a 28-bp DNA containing 1× ATF1DBD binding site as the specific binding sequence, and a scrambled 25-bp DNA as the controls sequence. **(c-d)** *In vitro* droplet experiments for: (c) 8 μM GFP-EWS-ATF1; (d) 8 μM GFP-EWS-ATF1 mixed with 0.36 μM 211-bp dsDNA. DNA contained 11× ATF1DBD binding sites and labeled with Quasar 670. The working buffer used for the *in vitro* droplet assays was 40 mM Tris-HCl (pH 7.5), 150 mM KCl, and 2 mM MgCl₂, 1 mM DTT, and 0.2 mg/ml BSA. The working buffer was the same as the working buffer for DNA Curtains and EMSA. **(e)** Dual-luciferase assays were performed 24 h after transfection of 293T cells with overexpression of EWS-ATF1 and a firefly luciferase vector possessing the indicated number of consecutive binding motifs preceding a minimal promoter. Data are normalized to relative luciferase activity of empty vector condition. Error bars represent s.d. (N = 3). The red dashed line represents Hill function fitting. Error bars of the fitting parameters represent 95 % confidence intervals obtained through Hill function fitting. **(f)** Wide-field TIRFM images of DNA Curtains after the 1st 10-min incubation of GFP-EWS-ATF1, like in Fig. 3b. flow on (i); flow off (ii). **(g)** Wide-field TIRFM images of GFP-EWS-ATF1 condensates and Pol II CTD_{N26}-mCherry after the 2nd 10-min incubation, like Fig. 3h. (i)-(ii) only the 488-nm laser on and flow on (i) / off (ii); (iii)-(iv) only the 561-nm laser on and flow on (iii) / off (iv); White Arrows pointed to puncta of Pol II CTD_{N26}-mCherry colocalized with GFP-EWS-ATF1 condensates. **(h)** Schematic of *in vitro* single-molecule biomolecular condensate-induced transcription assay for GFP-EWS-ATF1 (i). Wide-field TIRFM images of nascent RNA transcripts (iii) colocalized with EWS-FLI1 condensates (ii) after incubation. White arrows in (iii) confirmed the labeled punctum was on DNA. The working buffer contained 25 mM KCl.

Supplementary video 5 does not support the conclusion that EWS/FLI functions like FUS-Gal4. Fewer Pol II CTD puncta are observed and they tend to be much lower on the DNA strand than a majority of the areas of high YOYO-1 intensity suggesting EWS/FLI is either not phase separating at the GGAA microsatellite and that Pol II CTD is not recruited to the non-microsatellite DNA where EWS/FLI is phase separating.

We thank the reviewer for this question. We noticed that the low Pol II CTD recruitment efficiency by EWS-FLI1 came from the salt concentration, thus conducted new DNA Curtains experiments (Supplementary Fig. 6). This efficiency increased when the salt changed from 150 mM KCl (Supplementary Fig. 6b and d) to 25 mM KCl (Supplementary Fig. 6a and d). A new Supplementary video 5 in 25 mM KCl was prepared. Thus, EWS-FLI1 condensates can show robust Pol II recruitment to EWS/FLI-bound loci.

For the FUS-Gal4 experiments where phase separation is said to be mechanistically important (i.e. Pol II CTD recruitment and transcription initiation) the authors should directly test that by mutating the (G/S)Y(G/S) tyrosine residues thought to be critical for phase separation in FET proteins. This is actually a nice assay to do this, because unlike for EWS/FLI, disrupting the phase separation phenotype shouldn't impact the DNA binding activity of the construct.

We thank the reviewer for this suggestion. We followed the previous^{9,17} references to mutate all Tyrosine residues in 27 [G/S]Y[G/S] tripeptide repeats of FUS LCD to Serine residues, and this mutant was named as GFP-FUS(Y-S)-Gal4. First, GFP-FUS(Y-S)-Gal4 have much stronger binding to the 1× Gal4DBD sequence compared to the control sequence, suggesting a more specific interaction between FUS-Gal4 and

Gal4DBD (Response Fig. 3.4a). As the reviewer mentioned, disrupting the phase separation phenotype cannot impact the DNA binding activity of GFP-FUS(Y-S)-Gal4. In comparison to GFP-FUS-Gal4 (Response Fig. 3.4b or Fig.1a), GFP-FUS(Y-S)-Gal4 cannot form droplet (Response Fig. 3.4c), suggesting that the (G/S)Y(G/S) tyrosine residues are critical for phase separation in FET proteins, as the reviewer mentioned.

Response Fig. 3.4 | Molecular grammar of phase separation for FET fusion proteins. (a) GFP-FUS(Y-S)-Gal4. (i) Schematic; (ii) EMSA (12% native PAGE gel). We chose a 25-bp DNA containing 1x Gal4DBD binding site as the specific binding sequence, and a scrambled 25-bp DNA as the controls sequence. The working buffer

included 40 mM Tris-HCl (pH 7.5), 150 mM KCl, 2 mM MgCl₂, 1 mM DTT and 0.2 mg/ml BSA. The working buffer for EMSA was the same as the working buffer for DNA Curtains experiments. **(b-c)** *In vitro* droplet experiments for GFP-FUS-Gal4 (a), GFP-FUS(Y-S)-Gal4 (b). (i) 2 μM; (ii) 5 μM; (iii) 10 μM; (iv) 20 μM; (v) 30 μM. (a) is the same data of Fig. 1a.

While the luciferase assays in Figure 5 show similar activity thresholds as those previously identified by Gangwal et al and Johnson et al, similar concerns are present regarding the ability to compare FUS-Gal4 to EWS/FLI in luciferase assays. Just looking at the scale of the induced luminescence suggests the difference in DNA binding mechanism of ETS factors and Gal4 may play a role in assay function.

We thank the reviewer for this suggestion. We totally agree with the reviewer's comments, and the difference in DNA binding mechanism of ETS factors and Gal4 may play a role in assay function. Thus, we performed new DNA Curtains experiments to directly study EWS-FLI1. We found that EWS-FLI1 can recruit Pol II CTD (Supplementary Fig. 6a and d, and Supplementary video S5), and the recruitment of Pol II CTD by EWS-FLI1 condensates can activate gene transcription around the target loci (Fig. 4g). As suggested by the reviewer above, we also showed that another real FET fusion protein, EWS-ATF1 has the same result as FUS-Gal4 above. Especially for the luciferase assays (Response Fig. 3.3e), the scale of the induced luminescence of EWS-ATF1 is similar as FUS-Gal4. Here ATF1DBD can dimerize to bind DNA, like Gal4DBD. Taken together, although FLI1DBD and Gal4DBD/ATF1DBD were different, EWS-FLI1, EWSATF1, and our model system FUS-Gal4 showed the same behaviors: all of them can form biomolecular condensates on DNA, and the

condensates can recruit Pol II CTD to activate gene transcription around the target loci.

Figures 5d and 5e are basically the same analyses as was performed by Johnson et al. 2017 (PLoS ONE). The way the paper is currently written suggests that the identification of a threshold is a new contribution to the field, but this idea of a threshold has been previously shown by Gangwal et al (2008 and 2010) and Johnson et al (2017 PNAS and PLoS ONE). The authors add the analysis with the total motif and this particular addition should be highlighted as further supporting previous work, rather than as a novel finding.

We thank the reviewer for this suggestion. We totally agree with the reviewer, and moved the ChIP-Seq data from Fig. 5 to the supplementary part (Supplementary Fig. 8). We revised the relevant main text at Page 15, and also cited the relevant references suggested by the reviewer.

For Figure 6, please include the data for the 3X, 7X, 9X, and 13X GGAA repeats. This is needed to draw the conclusion that “a threshold number of GGAA repeats is required” as stated in the text.

We thank the reviewer for this comment. As suggested, we updated Fig. 6.

Figure 6 is impacted by the same issues as Figure 2.

Specifically that mCherry-EWS/FLI binds at many other regions than just the GGAA microsatellite. This undermines the purpose of the figure and suggest that consecutive

GGAA repeats do not contribute to EWS/FLI binding at DNA. The authors attempt to explain this by showing that there is a threshold of the total number of GGAA repeats in a stretch of DNA rather than the consecutive motifs. However, in their analysis of the lambda DNA in Figure 6, they switch the definition from motifs within 20-bp (as previously published and from the analysis for Figure 5) to motifs within 160-bp. This is a much larger window and it is unclear that this holds up in a genome-wide analysis. They should keep the window for total motifs consistent between Figure 5 and Figure 6.

We thank the reviewer for this comment. As suggested, we updated Fig. 6 to keep the window for total motifs consistent between Fig. 5 and Fig. 6 (within 20-bp).

Together, the explanation that it is just simply the total number of GGAA motifs that is important for phase separation and Pol II recruitment, but not consecutive motifs, is contrary to much of the evidence in the field to date. It may be true in vitro, but the in vivo evidence suggests that consecutive motifs carry real biological significance. Consecutive motifs are more highly bound by EWS/FLI (ChIP-seq) and the change of a single base pair to generate a new string of consecutive motifs (Grünwald et al. Nat Genet 2016). Consider revising discussion of total vs. consecutive motifs with this in mind.

We thank the reviewer for these great suggestions. Here, we believe that the total motifs in Fig. 6 can only be better to explain the biomolecular condensates formation on DNA Curtains, and it is still not clear whether the total motifs are important for Pol II recruitment or not. We totally agree with the referee's comments about the consecutive motifs. Although it seems that the total motifs can be used to explain the biomolecular condensates formation on DNA Curtains, we still believe that the

consecutive motifs carry real biological significance. We toned down the statement in our manuscript and reviewed the relevant main text at Page 20 accordingly. We also cited the relevant references mentioned by the referee.

In the first paragraph of the discussion, the authors state that the “transcription assay confirms that recruited Pol II molecules indeed activate gene transcription, and hence possibly exerts aberrant transactivation in cancer patient cells.” This is a bit of an overstatement, as the in vitro assay used an artificial construct for both the DNA-binder and Pol II. Additionally, the Supplemental Video 5 doesn’t show robust Pol II recruitment to EWS/FLI-bound loci.

We thank the reviewer for this comment. We conducted new experiments with a genuine construct EWS-FLI1/EWS-ATF1. We found that EWS-FLI1/EWS-ATF1 can recruit Pol II CTD (Supplementary Fig. 6a and d, new Supplementary video S5, and Response Fig. 3.3g) with high efficiency, and the recruitment of Pol II CTD by EWS-FLI1/EWS-ATF1 condensates can activate gene transcription around the target loci under 25 mM KCl (Fig. 4g and Response Fig. 3.3h).

Taken together, on the basis of these new experiments, we think the statement mentioned by the reviewer should be OK.

Minor concerns:

The first sentence refers to “lipid membranes-separated,” in this phrase, the membrane should not be plural. Find and replace instances of this to “lipid membrane-separated” throughout the paper.

We thank the reviewer for this suggestion. As suggested, we revised the relevant main text at Page 3.

The current nomenclature is “Ewing sarcoma,” not “Ewing’s sarcoma.” Be sure to consistently use the correct nomenclature throughout the paper.

We thank the reviewer for this suggestion. As suggested, we revised the relevant main text at Page 3, 6, 14, and 34.

For Figure 1k, where are the pre-bleach intensities for the DNA-containing samples?

We thank the reviewer for this suggestion. All three curves overlapping together caused this problem. We fixed this problem by adjusting the transparencies of the navy curve (samples without DNA) and purple curve (samples with DNA containing 1× GGAA repeat). We revised this figure accordingly.

Page 6, paragraph 1: the second sentence needs to be re-written and likely split into two sentences to better communicate the rationale for using 25x GGAA microsatellites.

We thank the reviewer for this suggestion. As suggested, we revised the relevant main text at Page 6.

YOYO-1 staining needs to be introduced for readers unfamiliar with it before the discussion of Figure 2.

We thank the reviewer for this suggestion. As suggested, we revised the relevant main text at Page 6.

On page 13 the second to last line reads “to record the distribution of GGAA microsatellites in the A673 genome.” This suggests to the reader that the authors have reanalyzed the ChIP-seq data with a custom A673 genome, not hg19 as discussed in the methods. This should just say “to record the distribution of GGAA microsatellites in the genome.”

We thank the reviewer for this suggestion. The reviewer is correct. Hg19 was used as the reference genome for sequence read mapping. We have revised the relevant main text at Page 15.

For Supplementary Figure 5, the y-axes should all be set equally.

We thank the reviewer for this suggestion. As suggested, we revised this figure.

On page 17, in the second paragraph the authors state that “not only LCD-LCD interactions, but also LCD-DBD interactions” are important has already been shown by Johnson et al (2017 PNAS). This reference should be added.

We thank the reviewer for this suggestion. As suggested, we revised the relevant main text and added the reference at Page 19.

The file for Supplementary Video 3 is broken. I tried downloading it in the zip for reviewer and the source data, but it never worked.

We thank the reviewer for this question. As suggested, we prepared the new video file.

References

- 1 Johnson, K. M. *et al.* Role for the EWS domain of EWS/FLI in binding GGAA-microsatellites required for Ewing sarcoma anchorage independent growth. *Proc. Natl. Acad. Sci. U. S. A.* **114**, 9870-9875, doi:10.1073/pnas.1701872114 (2017).
- 2 Johnson, K. M., Taslim, C., Saund, R. S. & Lessnick, S. L. Identification of two types of GGAA-microsatellites and their roles in EWS/FLI binding and gene regulation in Ewing sarcoma. *PLoS One* **12**, doi:ARTN e0186275, 10.1371/journal.pone.0186275 (2017).
- 3 Monument, M. J. *et al.* Clinical and Biochemical Function of Polymorphic NR0B1 GGAA-Microsatellites in Ewing Sarcoma: A Report from the Children's Oncology Group. *PLoS One* **9**, doi:ARTN e104378, 10.1371/journal.pone.0104378 (2014).
- 4 Gangwal, K. *et al.* Microsatellites as EWS/FLI response elements in Ewing's sarcoma. *Proc. Natl. Acad. Sci. U. S. A.* **105**, 10149-10154, doi:10.1073/pnas.0801073105 (2008).
- 5 Sizemore, G. M., Pitarresi, J. R., Balakrishnan, S. & Ostrowski, M. C. The ETS family of oncogenic transcription factors in solid tumours. *Nature Reviews Cancer* **17**, 337-351, doi:10.1038/nrc.2017.20 (2017).
- 6 Uren, A., Tcherkasskaya, O. & Toretsky, J. A. Recombinant EWS-FLI1 oncoprotein activates transcription. *Biochemistry* **43**, 13579-13589, doi:10.1021/bi048776q (2004).
- 7 Thomsen, C., Grundevik, P., Elias, P., Stahlberg, A. & Aman, P. A conserved N-terminal motif is required for complex formation between FUS, EWSR1, TAF15 and their oncogenic fusion proteins. *FASEB J.* **27**, 4965-4974, doi:10.1096/fj.13-234435 (2013).
- 8 Burke, K. A., Janke, A. M., Rhine, C. L. & Fawzi, N. L. Residue-by-Residue View of In Vitro FUS Granules that Bind the C-Terminal Domain of RNA Polymerase II. *Mol. Cell* **60**, 231-241, doi:10.1016/j.molcel.2015.09.006 (2015).
- 9 Wang, J. *et al.* A Molecular Grammar Governing the Driving Forces for Phase Separation of Prion-like RNA Binding Proteins. *Cell* **174**, 688-+, doi:10.1016/j.cell.2018.06.006 (2018).
- 10 Boija, A. *et al.* Transcription Factors Activate Genes through the Phase-Separation Capacity of Their Activation Domains. *Cell* **175**, 1842-+, doi:10.1016/j.cell.2018.10.042 (2018).
- 11 Sabari, B. R. *et al.* Coactivator condensation at super-enhancers links phase separation and gene control. *Science* **361**, 379-+, doi:10.1126/science.aar3958;eaar3958 (2018).
- 12 Wang, L. *et al.* Histone Modifications Regulate Chromatin Compartmentalization by Contributing to a Phase Separation Mechanism. *Mol. Cell* **76**, 646-+, doi:10.1016/j.molcel.2019.08.019 (2019).
- 13 Wang, L. *et al.* Rett syndrome-causing mutations compromise MeCP2-mediated liquid-liquid phase separation of chromatin. *Cell Res.* **30**, 393-407, doi:10.1038/s41422-020-0288-7 (2020).
- 14 Fang, X. F. *et al.* Arabidopsis FLL2 promotes liquid-liquid phase separation of polyadenylation complexes. *Nature* **569**, 265-+, doi:10.1038/s41586-019-1165-8 (2019).
- 15 Greene, E. C., Wind, S., Fazio, T., Gorman, J. & Visnapuu, M. L. in *Methods in Enzymology, Vol 472: Single Molecule Tools, Pt A: Fluorescence Based*

- Approaches* Vol. 472 *Methods in Enzymology* (ed N. G. Walter) 293-315 (Elsevier Academic Press Inc, 2010).
- 16 Zhao, Y. L., Jiang, Y. Z. & Qi, Z. Visualizing biological reaction intermediates with DNA curtains. *J. Phys. D-Appl. Phys.* **50**, 16, doi:10.1088/1361-6463/aa59cf (2017).
- 17 Kato, M. *et al.* Cell-free Formation of RNA Granules: Low Complexity Sequence Domains Form Dynamic Fibers within Hydrogels. *Cell* **149**, 753-767, doi:10.1016/j.cell.2012.04.017 (2012).
- 18 Gorthi, A. *et al.* EWS-FLI1 increases transcription to cause R-loops and block BRCA1 repair in Ewing sarcoma. *Nature* **555**, 387+, doi:UNSP 25748 10.1038/nature25748 (2018).
- 19 Qi, Z. *et al.* DNA Sequence Alignment by Microhomology Sampling during Homologous Recombination. *Cell* **160**, 856-869, doi:10.1016/j.cell.2015.01.029 (2015).
- 20 Pedelacq, J. D., Cabantous, S., Tran, T., Terwilliger, T. C. & Waldo, G. S. Engineering and characterization of a superfolder green fluorescent protein (vol 24, pg 79, 2005). *Nat. Biotechnol.* **24**, 1170-1170, doi:10.1038/nbt0906-1170d (2006).

REVIEWERS' COMMENTS

Reviewer #1 (Remarks to the Author):

The authors addressed most of the concerns we raised. The paper is strengthened and improved by the added control experiment.

Reviewer #2 (Remarks to the Author):

The authors have addressed all the issues that I raised. For my part, I have no further requests of the authors. This is a very interesting manuscript and the extent of new data provided here go above and beyond what should be considered satisfactory. In my view, this version of the MS is indeed suitable for publication.

Reviewer #3 (Remarks to the Author):

On the whole Zuo et al. have adequately addressed this reviewer's concerns and this manuscript is acceptable for publication in its revised form.

One minor concern is that EWS/FLI doesn't recruit PolII in 150 mM salt, and instead required 25 mM salt. This doesn't necessarily represent physiologically-relevant conditions, however, these are highly artificial conditions to begin with.

Final revisions for Nature Communications manuscript NCOMMS-20-21264A

Feb. 3, 2021

REVIEWERS' COMMENTS

Reviewer #1 (Remarks to the Author):

The authors addressed most of the concerns we raised. The paper is strengthened and improved by the added control experiment.

We thank the reviewer's positive comments.

Reviewer #2 (Remarks to the Author):

The authors have addressed all the issues that I raised. For my part, I have no further requests of the authors. This is a very interesting manuscript and the extent of new data provided here go above and beyond what should be considered satisfactory. In my view, this version of the MS is indeed suitable for publication.

We thank the reviewer's positive comments.

Reviewer #3 (Remarks to the Author):

On the whole Zuo et al. have adequately addressed this reviewer's concerns and this manuscript is acceptable for publication in its revised form.

One minor concern is that EWS/FLI doesn't recruit PolII in 150 mM salt, and instead required 25 mM salt. This doesn't necessarily represent physiologically-relevant conditions, however, these are highly artificial conditions to begin with.

We thank the reviewer's positive comments. We appreciate the reviewer's concern, and this is an interesting point warrant future studies.